# Efficient intracellular delivery of proteins by a multifunctional chimaeric peptide in vitro and in vivo

Siyuan Yu[1,2], Han Yang[1,2], Tingdong Li[1,2], Haifeng Pan[1,2], Shuling Ren[1], Guoxing Luo[1], Jinlu Jiang[1], Linqi Yu[1], Binbing Chen[1], Yali Zhang[1], Shaojuan Wang[1], Rui Tian[1], Tianying Zhang [1], Shiyin Zhang[1], Yixin Chen [1], Quan Yuan [1✉], Shengxiang Ge [1✉], Jun Zhang[1] & Ningshao Xia [1✉]

Protein delivery with cell-penetrating peptide is opening up the possibility of using targets inside cells for therapeutic or biological applications; however, cell-penetrating peptide-mediated protein delivery commonly suffers from ineffective endosomal escape and low tolerance in serum, thereby limiting in vivo efficacy. Here, we present an intracellular protein delivery system consisting of four modules in series: cell-penetrating peptide, pH-dependent membrane active peptide, endosome-specific protease sites and a leucine zipper. This system exhibits enhanced delivery efficiency and serum tolerance, depending on proteolytic cleavage-facilitated endosomal escape and leucine zipper-based dimerisation. Intravenous injection of protein phosphatase 1B fused with this system successfully suppresses the tumour necrosis factor-α-induced systemic inflammatory response and acetaminophen-induced acute liver failure in a mouse model. We believe that the strategy of using multifunctional chimaeric peptides is valuable for the development of cell-penetrating peptide-based protein delivery systems, and facilitate the development of biological macromolecular drugs for use against intracellular targets.

[1] State Key Laboratory of Molecular Vaccinology and Molecular Diagnostics, National Institute of Diagnostics and Vaccine Development in Infectious Diseases, Collaborative Innovation Centers of Biological Products, School of Public Health, Xiamen University, Xiamen, China. [2]These authors contributed equally: Siyuan Yu, Han Yang, Tingdong Li, Haifeng Pan. ✉email: yuanquan@xmu.edu.cn; sxge@xmu.edu.cn; nsxia@xmu.edu.cn

Development of bio-macromolecular drugs become a trend in the expansion of drugs, and these drugs have the advantages of high activity and specificity[1], low toxicity[2] and a wide range of target molecules[3]. Compared to genetic-manipulating drugs, protein-based therapeutics show faster action, greater controllability of the functional intensity and duration, and no genetic toxicity[4]. However, the application of protein therapeutics is limited by the poor cell membrane permeability of the agents. One method for protein delivery into cells involves using cell-penetrating peptides (CPPs), which can deliver protein cargos into cells through a non-invasive route and has many appealing applications both in vitro and in vivo[5]. Several CPP-based therapeutics have been entered into phase III clinical trials[6].

However, critical challenges have hindered CPP-based intracellular delivery. The principal challenge involves endosomal entrapment. After endocytosis is mediated by CPPs, the majority of the cargos (e.g. ~99% in the delivery of Cre recombinase)[7] are entrapped inside endosomes and eventually targeted and delivered to the lysosome for degradation[8,9]. Although pH-dependent membrane active peptides (PMAPs) are employed to disrupt the endosomal membrane, the efficiency of endosomal escape remains low[10]. This outcome may be due to the interaction between CPP or PMAP and the endosomal membrane. Accordingly, the efficient endosomal escape of fused cargos can be realised by removing fused cargo from CPP-PMAP in endosomes. Another challenge involves low serum tolerance, evident by sharply decreased delivery efficiency in the presence of high concentrations of serum[11], which severely limits its application in situations in which serum exposure is unavoidable, particularly for in vivo applications. It has been previously speculated that the suppressed cellular uptake is mainly caused by the binding of CPPs to negatively charged molecules in serum, such as albumin[12,13]. A multivalent CPP strategy[14], such as use of a branched peptide system[15] and attaching a protein oligomerization domain (e.g. p53tet)[16] to CPPs, may increase the local concentration of peptides that can actually interact with cellular membrane components, which may reduce the competitiveness of albumin in serum.

With a fundamental understanding of these issues, we designed a strategy of multifunctional chimaeric peptide development that is based on a rational design with inclusion of three additional modules: PMAP, cleavage sites recognised by endosomal proteases, and a leucine zipper with the capacity for homodimerization, to overcome the existing drawbacks of the original CPP system. We demonstrate that eTAT exhibits enhanced delivery efficacy both in vivo and in vitro, which is attributed to the function of the supplemental modules, especially proteolytic cleavage-facilitated endosomal escape, as well as dimerisation-mediated enhanced endocytosis and serum tolerance. We further show that intravenous injection of eTAT-protein phosphatase 1B (Ppm1b) successfully suppresses the tumour necrosis factor-α-induced systemic inflammatory response and cures acetaminophen-induced acute liver failure in a mouse model. Therefore, we believe that the eTAT system or an improved version of it composed of multifunctional chimaeric peptides has the potential to become a major strategy enabling protein delivery in biological research and therapeutic applications.

## Results

### Proteolytic removal of CPP-PMAPs in endosomes promotes the escape of protein cargos

The low levels of endosomal escape of cargo delivered by the CPP-PMAP system are possibly due to the cargo being tethered to the endosomal membrane via the interaction between CPP-PMAP and endosomal membrane components[10]. We speculate that removal of CPP-PMAP from the fused cargo via proteolytic cleavage in the endosomes promotes endosomal escape of the cargo. To test this hypothesis, a series of recombinant proteins were designed and prepared with TAT (T)[17] and INF7 (I)[18] serving as a CPP and PMAP, respectively, and different cleavage sites recognised by endosome-localised proteases, such as cathepsin L (CTSL)[19–21], cathepsin D (CTSD)[22] and furin[23,24], were inserted between the TAT-INF7 (TI) sequence and cargo (Fig. 1a). To better quantify the endosomal escape efficiency mediated by different constructs, we established an endonuclear split green fluorescent protein (GFP) reporter system (Fig. 1b). In this system, $GFP_{11}$ was fused to mRuby3 (red fluorescent protein) and histone H3 constitutively to allow its overexpression and visualisation in the nucleus of HEK-293T cells (HEK-293T-$GFP_{11}$) (Fig. 1b, Supplementary Fig. 1a), and cells expressing mRuby3 at high levels were identified and selected to evaluate the delivery efficiency of different constructs. After screening, 99.6% of living HEK-293T-$GFP_{11}$ cells expressed high levels of mRuby3 (Supplementary Fig. 1b), as determined by fluorescence activated cell sorter (FACS). The nonfluorescent $GFP_{1-10}$ fragment ($GFP_{1-10}$)[25] with a nuclear location signal (NLS)[26] fused to its C-terminus ($GFP_{1-10}$-NLS) was used as the cargo (Fig. 1a, Supplementary Fig. 2a). The relative mean fluorescence intensity (MFI) (the fluorescence intensity of total cells treated with different $GFP_{1-10}$ constructs compared to that of all cells treated with $GFP_{1-10}$ alone) (Fig. 1b) was used as the main index of the average amounts of $GFP_{1-10}$ delivered to each cell.

Our data demonstrated that the introduction of proteolytic sites resulted in the improvement of endosomal escape, and the proteins harbouring cathepsin L cleavage site N (TIN-$GFP_{1-10}$-NLS) and furin cleavage site Ne (TINe-$GFP_{1-10}$-NLS) had the highest MFI (Fig. 1c). When both N and Ne sites were simultaneously introduced (TINNe-$GFP_{1-10}$-NLS) (Supplementary Fig. 2b), endosomal escape was further enhanced (Fig. 1d and Supplementary Fig. 3). To investigate the relationship between proteolytic cleavage and endosomal escape, the endocytosed proteins were analysed by western blotting, and the MFI of the cells was monitored during the endonuclear split-GFP assay (Fig. 1e, f). Removal of TAT-INF7 via proteolytic cleavage in endosomes significantly alleviated the degradation of $GFP_{1-10}$ and ultimately resulted in a higher MFI in the treated cells. The arginine-to-glycine mutation in the N and Ne sites[27,28] abrogated protein cleavage in endosomes, and the endosomal escape efficiency was decreased compared to that of proteins harbouring wild-type N and Ne sites (Supplementary Fig. 4). In addition, disruption of the endosomal membrane by PMAP was a prerequisite for the enhanced endosomal escape obtained via proteolytic cleavage. In the absence of INF7, TNNe-$GFP_{1-10}$-NLS was cleaved in the endosomes to an extent similar to the cleavage of TINNe-$GFP_{1-10}$-NLS, but the former was still entrapped in the endosomes and rapidly degraded, similar to its uncleaved counterparts (Supplementary Fig. 5). Notably, poly-histidine tag (6 × His-tag) was presented in all of $GFP_{1-10}$-NLS-related recombinant proteins to facilitate the purification of these proteins. Histidine residues are known to serve as a proton sponge and facilitate escape from endosomes, thus poly-histidine sequences have been used as motifs to improve endosomal escape in trans-delivery (co-incubation) of gene[29] or ribonucleoproteins[30] previously. To evaluate the impact of 6 × His-tag on endosomal escape when fused to CPP-cargo proteins, the HEK-293T-$GFP_{11}$ cells were treated by 5 μM TINNe-$GFP_{1-10}$-NLS with 6 × His-tag (6H+) or not (6H−) (Supplementary Fig. 6a, b) separately. Our data show that there is no significant difference between the MFI of HEK-293T-$GFP_{11}$ treated by the two proteins, suggesting that proton sponge effect of His-tag is not obvious when fused to CPP-cargo proteins (Supplementary Fig. 6c).

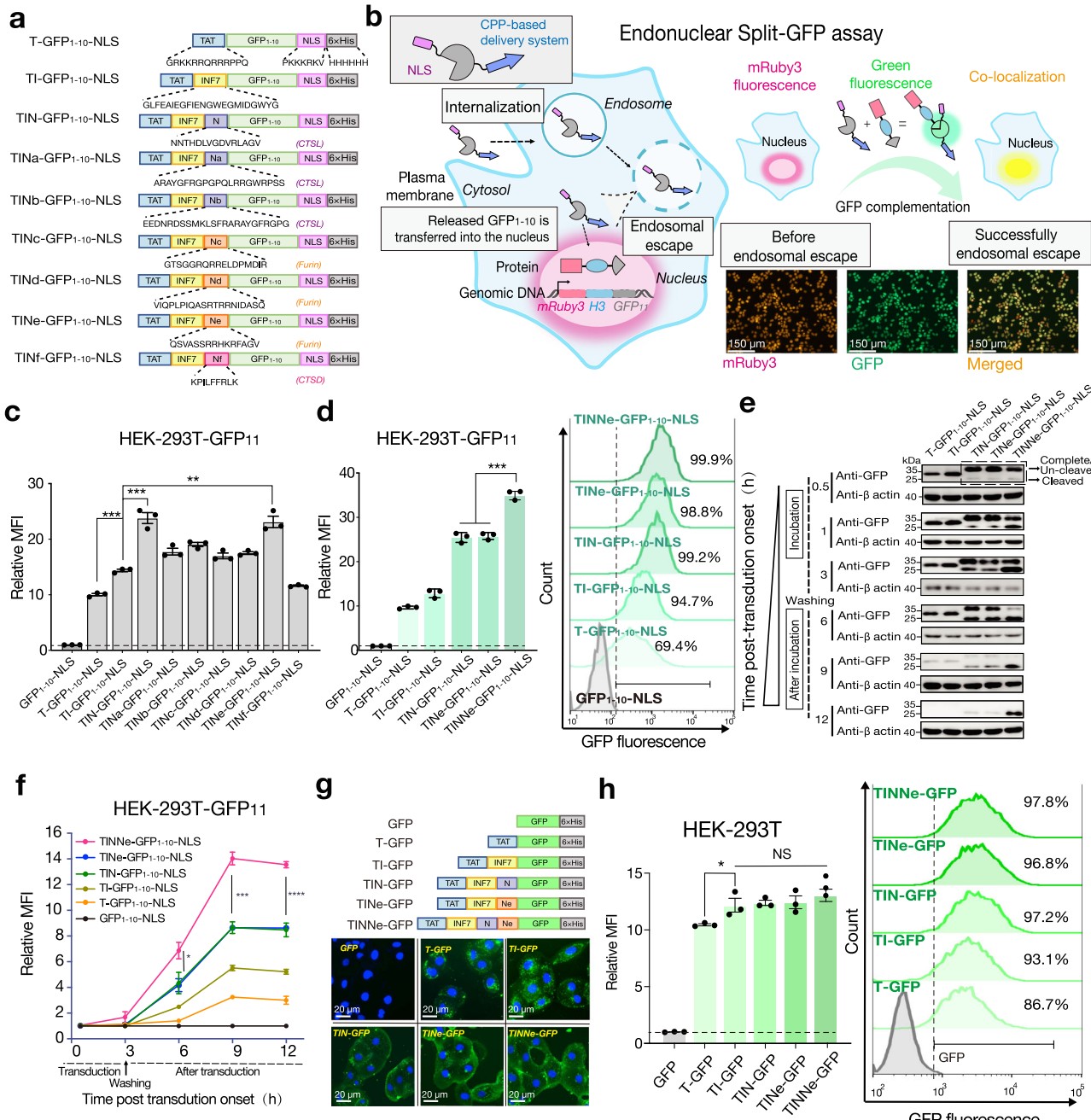

**Fig. 1 Proteolytic removal of CPP-PMAP in endosomes enhances the escape efficiency of protein cargos. a** Schematic diagram of different GFP$_{1–10}$-NLS-related recombinant proteins. The amino acid sequences of TAT, INF7, NLS, 6 × His and the different proteolytic cleavage sites (N and Na-Nf) are presented below, and each protease-recognised cleavage site is labelled in brackets. **b** Schematics of the endonuclear split GFP assay. Only after GFP$_{1–10}$-NLS was released from the endosome and translocated into the nucleus of HEK-293T cells with stable expression of histone H3-GFP β-sheet 11 (HEK-293T GFP$_{11}$), the fluorescence of the complete GFP in the nucleus observed, and co-localised with the mRuby3 fluorescence. **c** The MFI of HEK-293T GFP$_{11}$ cells treated with GFP$_{1–10}$-NLS-related proteins containing different cleavage sites (5 μM). **d** The MFI (left panel) within total HEK-293T cells and the corresponding percentage of green fluorescence-positive cells (right panel) treated with GFP$_{1–10}$-NLS-related proteins containing two cleavage sites (5 μM). **e** Immunoblot analysis of GFP and **f** the MFI in the treated cells at different time points during the endonuclear split GFP assay (1 μM). **g** Representative microscopy images of cytosolic fluorescence distribution in MA-104 cells treated with different GFP-related proteins (1 μM) as indicated (upper panel: scheme diagram of GFP-related constructs). **h** The MFI (left panel) within total HEK-293T cells and the corresponding percentage of green fluorescence-positive cells (right panel) after treatment with GFP-related proteins (1 μM). For (**c**, **d**, **f**, and **h**) the results shown are the means ± s.e.m.; $n = 3$ biologically independent samples; *$P < 0.05$, **$P < 0.01$, ***$P < 0.001$, and NS no significant difference; two-tailed unpaired student's $t$ test. Relative MFI (fold increase) was obtained by MFI of total cells treated with the indicated proteins divided by that of total cells treated with the corresponding cargo protein only. For (**b** bottom panel) and (**g** bottom panel), the data shown are representative of three independent experiments, respectively; for (**e**), the data shown are representative of two independent experiments. For data, statistics, exact $P$ values and uncropped images of the immunoblots, see Source Data File.

To validate the effects of proteolytic cleavage, the cytosolic delivery of protein was examined by observing the distribution of green fluorescence in MA-104 cells (Fig. 1g) treated by different GFP-related proteins (Supplementary Fig. 1c). After three hours of incubation, T- and TI-GFP exhibited punctate distribution consistent with GFP localisation within endosomes. However, TIN- and TINe-GFP both showed a diffuse fluorescence distribution in the cytoplasm of the MA-104 cells, indicating that cleaved GFP diffused into the cytosol after escape from endosomes. This diffusion phenomenon was obvious when the two sites N and Ne were combined (Fig. 1g). Finally, to verify whether the increased MFI was due to the enhancement of internalisation (initial step of intracellular delivery) induced by the added functional modules, the MFI of HEK-293T cells treated with this series of GFP-related proteins (Fig. 1h) was analysed by FACS. The results showed that only the addition of PMAP (INF7) slightly increased the internalisation efficiency (Fig. 1h). Collectively, these results clearly show that the proteolytic removal of CPP-PMAP from the cargo in endosomes promotes the endosomal escape of the cargo.

**Dimerisation of the CPP-fused protein enhances endocytosis and serum tolerance.** CPP-mediated intracellular delivery is often not satisfactory in the presence of serum, possibly because the electrostatic interaction between CPP and cell membrane components is competitively inhibited by negatively charged molecules in the serum, such as albumin[12,31]. However, no direct evidence was found to support this speculation. To evaluate whether negatively charged molecules in the serum can bind to CPP-fused proteins, GFP or T-GFP was mixed with bovine serum albumin (BSA) (molar ratio = 1:3) (Supplementary Fig. 7a) and then analysed by high pressure size exclusion chromatography (HPSEC). The HPSEC results showed that BSA can bind to T-GFP but not GFP (Supplementary Fig. 7b, c), indicating that BSA can bind specifically to the TAT motif of T-GFP. Moreover, to evaluate the impact of this binding on internalisation efficiency, HEK-293T cells were treated with T-GFP in the presence of 50 g/L BSA (albumin accounted for 35–50 g/L in the serum)[32] and analysed by FACS. The results demonstrated that TAT-mediated delivery was dramatically inhibited by ~61.3% or 100% in the presence of BSA or 100% FBS, respectively (Fig. 2a), suggesting that albumin appeared to be one of the most critical components associated with the serum intolerance observed in the TAT-mediated delivery system.

To evaluate the ability of MCPPs to overcome serum intolerance, leucine zipper (L) with homodimerization capacity[33] was employed to mediate the dimerisation of the CPP-fused proteins. T-GFP and its counterpart containing L (TL-GFP) were constructed and expressed (Fig. 2b). As analysed by SDS-PAGE and HPSEC, T-GFP and TL-GFP were found to be monomers and dimers, respectively (Fig. 2c). Multivalent CPP systems have been shown in a previous study to exhibit enhanced internalisation properties[34], TL-GFP also exhibited an ~3-fold higher MFI than T-GFP in the absence of serum when HEK-293T cells were treated with the two proteins at the same concentration (Fig. 2d). As the concentration of foetal bovine serum (FBS) in the medium was increased, the MFI in the cells treated with the two proteins decreased, but the inhibition of serum on TL-mediated endocytosis was weak compared to that of T-GFP (Fig. 2e). Even in the presence of 100% FBS, the MFI of TL-GFP-treated cells remained 70% that observed in the absence of FBS, while the MFI of T-GFP-treated cells decreased to a nearly undetectable level in the presence of >50% serum (Fig. 2e, f). These results clearly demonstrate that the dimerisation of TAT enhanced not only internalisation but also the serum tolerance of CPP-mediated

intracellular delivery of cargos. The negatively charged BSA can also bind to TL-GFP (Supplementary Fig. 7d); however, the MFI in HEK-293T cells treated with BSA and TL-GFP, similar to the cells incubated with 100% FBS (Fig. 2f). These results showed that the dimerisation of CPP-fused cargo exhibits serum tolerance, mainly by weakening the competitiveness of serum protein albumin.

To determine whether the enhanced internalisation conferred by the dimerisation of the TAT proteins results in more cargo being released from the endosomes in the absence of PMAP or NNe, T-GFP$_{1-10}$-NLS and TL-GFP$_{1-10}$-NLS were expressed. The retention times in HPSEC assays and the SDS-PAGE both suggested that the leucine zipper induced homogeneous dimer formation (Supplementary Fig. 8). T-GFP$_{1-10}$-NLS and TL-GFP$_{1-10}$-NLS were examined after incubation with HEK-293T GFP$_{11}$ to determine their endosomal escape efficiency; however, it was shown that there was no significant difference in the MFI between the T-GFP$_{1-10}$-NLS- and TL-GFP$_{1-10}$-NLS-treated cells; the MFIs for both were very low (Fig. 2d). Then, a leucine zipper was employed to dimerise TINNe (TINNeL) (Fig. 2a). SDS-PAGE and HPSEC analysis confirmed leucine zipper-induced dimerisation when added in combination with INF7 and NNe (Supplementary Fig. 9a). Dimerisation of TINNe (TINNeL) resulted in more effective internalisation of GFP than caused by its monomer counterpart (Fig. 2g). The diffuse fluorescence distribution in the cytoplasm of MA-104 cells indicates the role of proteasomal cleavage in TINNe- and TINNeL-mediated delivery. The MFI in the HEK-293T cells treated with TINNeL-GFP was ~4-fold greater than that of the HEK-293T cells treated with TINNe-GFP (Fig. 2h). When evaluated with the endonuclear split GFP assay (Supplementary Fig. 9b), TINNeL delivered more cargo into the cytosol than TINNe; accordingly, there was a significant difference observed in the MFIs of the cells treated with TINNeL-GFP$_{1-10}$-NLS and those treated with TINNe-GFP$_{1-10}$-NLS: treatment with TINNeL-GFP$_{1-10}$-NLS resulted in a significantly higher MFI than treatment with TINNe-GFP$_{1-10}$-NLS (Figs. 2i, j). Furthermore, upon the dimerisation of TAT, the dimerisation of TINNe (TINNeL) also resulted in improved serum tolerance. When the concentration of FBS was increased from 0 to 100% in the medium, the MFI in the TINNeL-GFP$_{1-10}$-NLS-treated cells decreased slightly but remained greater than 70%, whereas almost no MFI was observed in the TINNe-GFP$_{1-10}$-NLS-treated cells (Fig. 2k). As the delivery efficiency of CPP, in general, correlates with cytotoxicity, we performed a water-soluble tetrazolium-8 assay to evaluate the in vitro cytotoxicity induced by TINNeL-GFP and TINNe-GFP. We found no significant toxicity when INF7, NNe or the leucine zipper was introduced (Supplementary Fig. 10).

We further investigated the amount of cargo proteins internalised into cells and the fraction of cytosolic delivery (internalised cargo that reached the nucleus) by the TINNeL system. Cells were counted, harvested and lysed, and intracellular protein concentration were quantified (Supplementary Fig. 11a, c). After incubation with equimolar GFP-related proteins (1 μM), ~57.6 μM of GFP was present inside HEK-293T cells treated with TINNeL-GFP (Supplementary Fig. 11d). Consistent with the increase of MFI in the FACS analysis (Supplementary Fig. 11d), the amount of internalised cargo by the TINNeL system was ~4-fold higher than that of the TINNe and TAT peptide (one unit of relative MFI as analysed FACS indicates 1.4 μM GFP inside the cells) (Supplementary Fig. 12a). Similarly, the amount of cytosolic delivery (nucleus localisation in our assay) was quantified by measuring GFP$_{1-10}$-NLS in the nucleus. The amount of TINNeL mediated-cytosolic delivery was ~13-fold higher than that of the TAT peptide alone (Supplementary Fig. 11e), and the similar increase was observed in the FACS analysis (one unit of relative MFI as analysed by FACS indicates

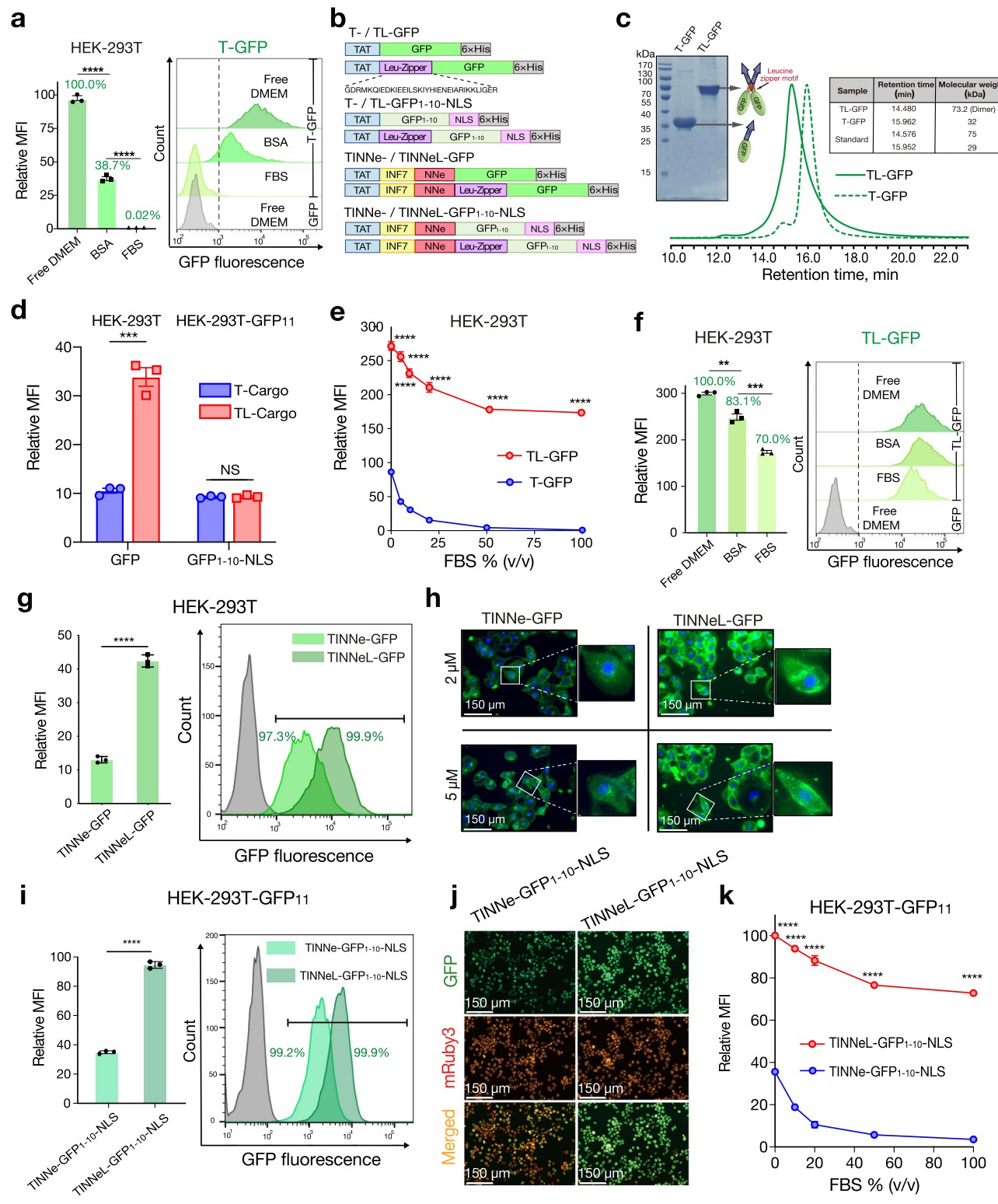

1.4 μM GFP$_{1-10}$-NLS inside the cells) (Supplementary Fig. 12b). Since the amount of internalisation between GFP and GFP$_{1-10}$-NLS was similar and mediated by the same delivery peptide (Supplementary Fig. 11f), we provide objective quantitative insights into the process of TINNeL-mediated endosomal escape, which suggest that TINNeL induced at least 30% of the total cellular internalisation, showing cargo escape from endosomes, remaining the same as that of TINNe, while only 10% of the internalised total it induced consisted of T-GFP$_{1-10}$-NLS (Supplementary Fig. 11g). Collectively,

these results show that the four-module chimaeric peptide TINNeL, which is named the enhanced TAT-based intracellular delivery system (eTAT), has the capacity to deliver cargos into the cytosol efficiently even in the presence of 100% serum.

**eTAT-Ppm1b suppresses TNF-induced necroptosis in vitro and in vivo.** Necroptosis can be induced by tumour necrosis factor-α (TNF-α), and the phosphorylation of

**Fig. 2 Dimerisation of CPP-fused proteins enhances endocytosis and serum tolerance. a** The MFI of T-GFP-treated HEK-293T cells (5 μM) in the indicated conditions (BSA, 50 g/L). **b** Schematic diagram of different GFP$_{1-10}$-NLS/GFP-related recombinant proteins. The amino acid sequence of the leucine zipper is presented below. **c** Reduced SDS-PAGE (protein samples not boiled) and HPSEC analysis of purified T- and TL-GFP; the table shows the retention time and molecular weight of samples. **d** The MFI of T- and TL-GFP-treated HEK-293T cells and MFI of the T- and TL-GFP$_{1-10}$-NLS-treated HEK-293T GFP$_{11}$ cells in free-serum DMEM. **e** The MFI of HEK-293T cells treated with 5 μM T- or TL-GFP in the presence of various concentrations of FBS. **f** The MFI of TL-GFP-treated HEK-293T cells (5 μM) in the indicated conditions (BSA, 50 g/L). **g** The MFI (left panel) within total HEK-293T cells and the corresponding percentage of green fluorescence-positive cells (right panel) after treatment with TINNe-GFP- or TINNeL-GFP. Proteins were used at a concentration of 1 μM. **h** Representative microscopy images of the cytosolic fluorescence distribution in MA-104 cells treated with TINNe-GFP and TINNeL-GFP at the indicated concentrations. **i** The MFI (left panel) of total HEK-293T-GFP$_{11}$ cells and the corresponding percentage of green fluorescence-positive cells (right panel) after treatment with TINNe-GFP$_{1-10}$-NLS or TINNeL-GFP$_{1-10}$-NLS at a concentration of 5 μM. **j** Representative microscope images of HEK-293T-GFP$_{11}$ cells treated with TINNe-GFP$_{1-10}$-NLS or TINNeL-GFP$_{1-10}$-NLS at concentrations of 5 μM. **k** The MFI of HEK-293T GFP$_{11}$ cells treated with TINNe-GFP$_{1-10}$-NLS or TINNeL-GFP$_{1-10}$-NLS in the presence of various concentrations of FBS. For (**a** (left panel), **d**– **f** (left panel), **g** (left panel), **i** (left panel) and **k**) the results shown represent the means ± s.e.m.; $n = 3$ biologically independent samples; *$P < 0.05$, **$P < 0.01$, ***$P < 0.001$, ****$P < 0.0001$ and NS no significant difference; two-tailed unpaired student's $t$ test. Relative MFI (fold increase) was obtained as above (Fig. 1). Percentages presented in the columns of (**a** and **f**) were obtained by dividing the MFI of treated cells in the given condition by the MFI of treated cells in free-serum DMEM. For (**c**, **h** and **j**) the data shown are representative of three independent experiments, respectively. For data, statistics, and exact $P$ values, see Source Data File.

receptor-interacting protein 3 (Rip3) in necrosomes is required to trigger necrosis[35]. Protein phosphatase 1B (Ppm1b) has been identified as a Rip3 phosphatase and has the ability to selectively suppress necroptosis through the dephosphorylation of Rip3[36]. In this study, Ppm1b was intracellularly delivered by different TAT-based methods (Supplementary Fig. 13) to suppress TNF-induced necroptosis in vitro and in vivo.

When evaluated using the L929 mouse fibroblast cell line, which is commonly used to study TNF-induced necroptosis[37], eTAT showed the highest efficiency for Ppm1b delivery, as analysed by western blotting at two different time points (Fig. 3a). The efficient cytosolic delivery of eTAT-Ppm1b at 12 h post incubation onset, was attributed to enhanced cellular internalisation and more efficient endosomal escape. The ability of this series of Ppm1b-related proteins to inhibit TNF-α-induced cell death was then explored. As shown in Fig. 3b and Supplementary Fig. 14 (Fig. 3b and Supplementary Fig. 14), the proportion of dead cells induced by mouse TNF-α and zVAD (TNZ) in the group pretreated with eTAT-Ppm1b was significantly lower than that in the groups pretreated with other recombinant Ppm1b proteins. Pretreatment with eTAT-Ppm1b provided even greater suppression of TNF-induced necroptosis than the positive control, in which Ppm1b was overexpressed through lentivirus infection (Lenti-Ppm1b); however, almost no suppression was observed in the T-Ppm1b- or Ppm1b-pretreated groups compared the negative control (PBS). Among the cells pretreated with Ppm1b proteins, the phosphorylation of Rip3 was lowest in the eTAT-Ppm1b-pretreated group (Fig. 3b).

TNF-induced necroptosis is essential for TNF-induced systemic inflammatory response syndrome (SIRS)[38], and the caecum is particularly sensitive to TNF-induced injury[36]. In this study, 5 nmol of different Ppm1b proteins was administered intravenously to prevent death, and 2 h later, the mice were challenged with a lethal dose of TNF injected into the tail vein[39] (Fig. 3c). TNF-induced death was completely prevented by eTAT-Ppm1b (100% survival) and partially prevented by TINNe-Ppm1b (50% survival), TI-Ppm1b (~17% survival) and T-Ppm1b (~17% survival) (Fig. 3c). TNF-induced caecal damage was significantly alleviated by pretreatment with eTAT-Ppm1b and TINNe-Ppm1b, and almost no histological change was observed in the caecum of the eTAT-Ppm1b-pretreated mice (Fig. 3d, e).

Generally, our results demonstrate that eTAT can deliver the functional protein Ppm1b and suppress TNF-induced necroptosis more efficiently in vitro and in vivo. To evaluate the delivery efficiency and tissue distribution of Ppm1b in vivo, Ppm1b, T-Ppm1b and eTAT-Ppm1b were labelled with the fluorescence dye Cy5, and the distribution of the fluorescence in wild-type

BALB/C mice was analysed using ex vivo imaging 24 h after intravenous administration. As expected, eTAT-Ppm1b showed higher fluorescence intensity but different extent than T-Ppm1b in all organs (Fig. 3f, g), confirming its enhanced delivery efficiency in vivo. Notably, eTAT-Ppm1b-Cy5, T-Ppm1b-Cy5 and Ppm1b-Cy5 were all mostly distributed in the caecum, followed by the liver, kidney, lung and spleen (Fig. 3f, g). Paraffin sections of caecal and liver tissues (two organs in which Ppm1b-related proteins were mainly distributed) were prepared, and the delivered Ppm1b was detected using anti-6 × His IgG as the primary antibody and Alexa 647-conjugated anti-IgG as the secondary antibody. The fluorescence images showed that a significant red fluorescence signal was observed in the caecum and liver of the eTAT-Ppm1b-treated mice, and the intensity was stronger than that of T-Ppm1b. At higher magnification, the Alexa-647 signal of eTAT-Ppm1b was bright and diffused in the cytosol of tissue cells, confirming in vivo intracellular protein delivery (Fig. 3h). To evaluate whether the caecal distribution was due to eTAT or Cy5, eTAT-GFP was labelled with Cy5 and injected into BALB/C mice. In comparison with eTAT-Ppm1b-Cy5, low fluorescence signal was observed in the caecum of the eTAT-GFP-Cy5-treated mice (Supplementary Fig. 15). The results demonstrate that Ppm1b itself, but not eTAT or any of its modules, induced the caecal distribution of eTAT cargo.

**eTAT-Ppm1b cures acetaminophen-induced acute liver failure.** Acetaminophen (APAP) is an effective analgesic and antipyretic drug when used at therapeutic levels, but acute liver failure (ALF) is induced by APAP an excessively high dose[40]. Although NAC, the only drug approved by the Food and Drug Administration of the United States for the treatment of APAP-induced ALF[41,42], attenuates APAP-induced hepatotoxicity, some patients may develop liver injury despite administration of the recommended dosage. Recently, Rip3 was found to be involved in the development of APAP-induced ALF[43]. In this study, Ppm1b was used as a treatment.

The maximum nonlethal and absolute minimum lethal doses of APAP injected intraperitoneally were 500 mg and 800 mg per kg body weight in overnight-fasted mice, respectively (Supplementary Fig. 16). For ALF therapy, overnight-fasted mice were intraperitoneally injected with a certain dose of APAP and were intravenously injected with 5 nmol Ppm1b protein twice: at 2 h and 6 h post-APAP injection (Fig. 4a, c). When acute liver damage was induced by the maximum nonlethal dose of APAP, treatment with eTAT-Ppm1b resulted in the greatest reduction in the elevated serum alanine aminotransferase (ALT) and aspartate

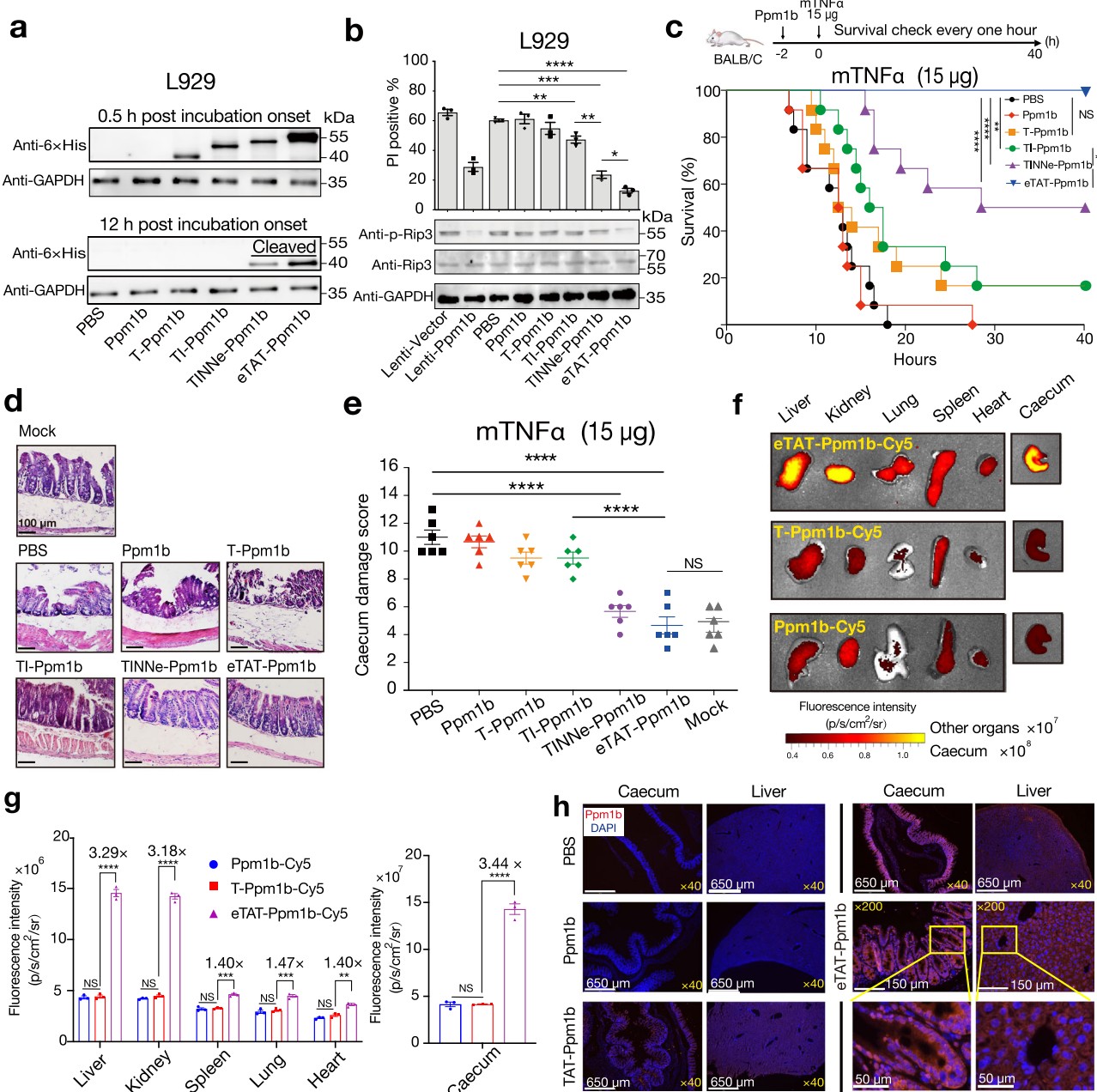

**Fig. 3 eTAT-Ppm1b suppresses TNF-induced necroptosis in vitro and in vivo. a** Immunoblot analysis of the level of Ppm1b in L929 cells treated with Ppm1b-related proteins (1 μM) at the indicated time points. ($n = 3$). **b** TNF-induced necroptosis of L929 cells pretreated with lentivirus harbouring the Ppm1b gene or Ppm1b-related recombinant proteins. Pretreated cells were stimulated with mTNF + zVAD for 5 h, washed and collected. The number of dead cells was analysed by flow cytometry using PI (upper panel), and the levels of Rip3 and phosphorylated Rip3 (p-Rip3) were analysed by immunoblotting using the indicated antibody (bottom panel); $n = 3$ biologically independent samples. **c–e** TNF challenge of female BALB/c mice pretreated intravenously with different Ppm1b-related recombinant proteins. The mouse survival curves after TNF challenge (**c**). Mouse survival was monitored every hour for 40 h with the results presented in a Kaplan–Meier plot, and a log-rank (Mantel-Cox) test was performed; *$P < 0.05$, **$P < 0.01$, ***$P < 0.001$, ****$P < 0.0001$, and NS no significant difference; $n = 12$ mice for each group pooled from two independent experiments. Histopathological appearance of the mouse caecum (**d**). Six hours post-TNF challenge, the caecum was collected, sectioned and stained with H&E. Representative images are shown, and the scale bar is 100 μm. Caecum damage score for the different groups (**e**). For (**d** and **e**) 6 mice from each group were analysed. PBS-pretreated mice served as a negative control, and mock mice were not challenged with TNF. **f, g** Distribution of intravenously administered Cy5 labelled-Ppm1b-related proteins in female BALB/c mice. Representative microscope images (**f**) and florescence intensity (**g**) in the indicated organs of BALB/c mice intravenously administered Ppm1b-Cy5, TAT-Ppm1b-Cy5 or eTAT-Ppm1b-Cy5 ($n = 3$). Twenty-four hours post-administration, the mice were sacrificed, and fluorescence imaging of each organ was performed. **h** Five hours after injection with the indicated constructs or PBS, the tissues were harvested and prepared as paraffin slides. The nuclei were stained with DAPI, and the delivered Ppm1b was detected using anti-6 × His IgG as the primary antibody and Alexa 647-conjugated anti-mouse IgG as the secondary antibody. Fluorescence was captured via fluorescence microscopy. Yellow boxes in the images indicate magnified regions; the data shown are representative of three independent experiments. The data in (**b** top panel, **e** and **g**) are expressed as the means ± s.e.m; two-tailed unpaired student's *t* test; *$P < 0.05$, **$P < 0.01$, ***$P < 0.001$, ****$P < 0.0001$, and NS no significant difference. For data, statistics, exact $P$ values, and uncropped images of the immunoblots, see Source Data File.

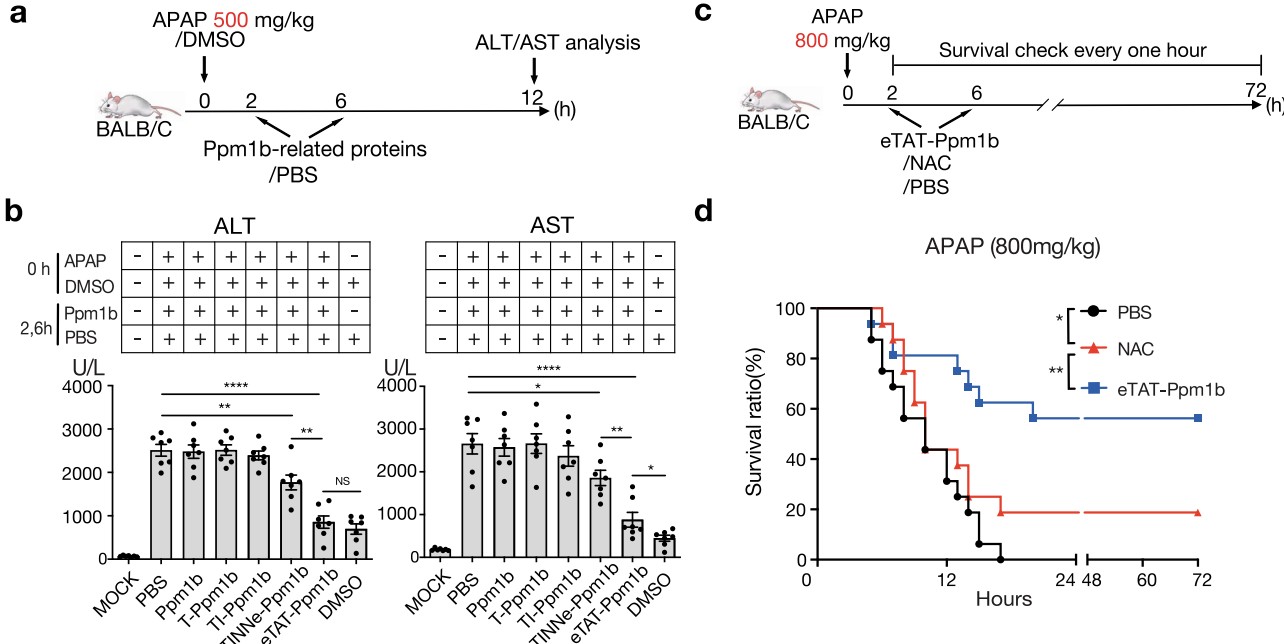

**Fig. 4 eTAT-Ppm1b cures acetaminophen-induced acute liver failure. a, b** eTAT-Ppm1b treatment of APAP-induced acute liver damage. Overnight-fasted female BALB/c mice were challenged intraperitoneally with APAP at a dose of 500 mg per kg body weight and then treated intravenously with Ppm1b-related recombinant proteins twice, 2 and 6 h after APAP administration (**a**). Twelve hours post-APAP challenge, serum was collected, and ALT and AST levels were determined (**b**). Mice treated with PBS were used as negative controls, and those challenged with DMSO solution and treated with PBS served as solvent controls. The mock mice received neither APAP nor Ppm1b-related proteins. All the values are expressed as the means ± s.e.m., $n = 7$ for each group; $*P < 0.05$, $**P < 0.01$, $***P < 0.001$, $****P < 0.0001$, and NS no significant difference; two-tailed unpaired student's $t$ test. **c, d** eTAT-Ppm1b therapy for attenuating APAP-induced death. Overnight-fasted female BALB/c mice were challenged intraperitoneally with APAP at a dose of 800 mg per kg body weight and then treated intravenously with Ppm1b-related recombinant proteins twice, 2 and 6 h after APAP administration (**c**). Then, the survival of the mice was monitored every hour for 72 h with the results presented in a Kaplan–Meier plot, and a log-rank (Mantel-Cox) test was performed; $*P < 0.05$, $**P < 0.01$ (**d**). Mice treated with PBS were used as negative controls; $n = 16$ mice for each group pooled from two independent experiments. For data, statistics, and exact $P$ values, see Source Data File.

aminotransferase (AST) levels, followed by decreases induced by TINNe-Ppm1b; however, TI-Ppm1b, T-Ppm1b and Ppm1b had no therapeutic effect (Fig. 4b). The therapeutic effect was achieved when two doses of 5 nmol (~15 mg/kg) eTAT-Ppm1b were administered (Supplementary Fig. 17). This effective dose was safe for the mice, as no significant change in ALT and AST in serum in our research was observed (Supplementary Fig. 18). This dose is moderate for CPP-based drugs via injection i.v., as indicated in a previous study (e.g. the median lethal dose of CPP-oligomer conjugates administered i.v. was between 210 and 250 mg/kg)[44,45]. Furthermore, 60% of the mice were rescued from death induced by administration of a minimum lethal dose of APAP by eTAT-Ppm1b treatment, but only 20% of the mice survived beyond the 72-h observation period when treated with N-acetylcysteine (NAC) at a dose of 600 mg per kg body weight (Fig. 4d). These results indicate that intravenous administration of eTAT-Ppm1b effectively alleviated APAP-induced acute liver damage and prevented APAP-induced death, thereby suggesting its valuable application potency in disease therapy.

## Discussion

Over the past three decades, CPPs have been extensively used as delivery tools for proteins in a plethora of cell types, as well as in preclinical models of disease and clinical trials; however, to date, no CPPs have been approved for sale[46], implying that the existing CPPs may show low efficacy in humans. Our study presents a strategy associated with multifunctional chimaeric peptides allowing efficient protein delivery in vivo and in vitro, which provides more possibilities for protein therapy.

The eTAT system presented here is composed of four functional modules: CPP (TAT), PAMP (INF7), combined with endosome-specific protease sites (NNe) and a leucine zipper for dimerisation (leucine zipper). As illustrated in Fig. 5, the endocytosis of cargo was mediated by the interaction between TAT and membrane components on the cell surface (e.g. glycosaminoglycans/GAGs) (Fig. 5a). The endosomal membrane was disrupted by the pH-dependent conformational changes of INF7 (Fig. 5b), and the cargo was separated from the membrane-bound TAT-INF7 through proteolytic cleavage of NNe by endogenous furin and CTSL in endosomes (Fig. 5c). Then, the dissociative cargo was rapidly released from the disrupted endosome and entered the cytosol (or nucleus) (Fig. 1). The noncovalent electrostatic interaction between the cargo and CPP-PMAP is very easy to dissociate competitively by negatively charged molecules in serum, while covalent linkers, such as disulfide bonds[10], are breakable only in the cytosol and may not result in highly efficient endosomal escape. Therefore, the cargo and CPP-PMAP should be covalently linked by a cleavable linker that can only be efficiently cleaved in endosomes. While the strategy of proteolytic cleavage activated-delivery has been utilised in the field of CPPs, after being pioneered by the group of Roger Y. Tsien[47,48], studies have focused on improving the specificity of CPP-based delivery[49]. For instance, proteolytic cleavage mediated by extracellular proteases (i.e. matrix metalloproteinases, MMPs) specifically restores the penetrating activity of the CPP moiety in the vicinity of tumour cells[47]. In the present study, we utilised endosome-localised proteases (e.g. CTSL and furin) to enhance endosomal escape. Besides, the enhanced efficiency of endosomal escape was proved (Fig. 1h) not due to the enhancement of

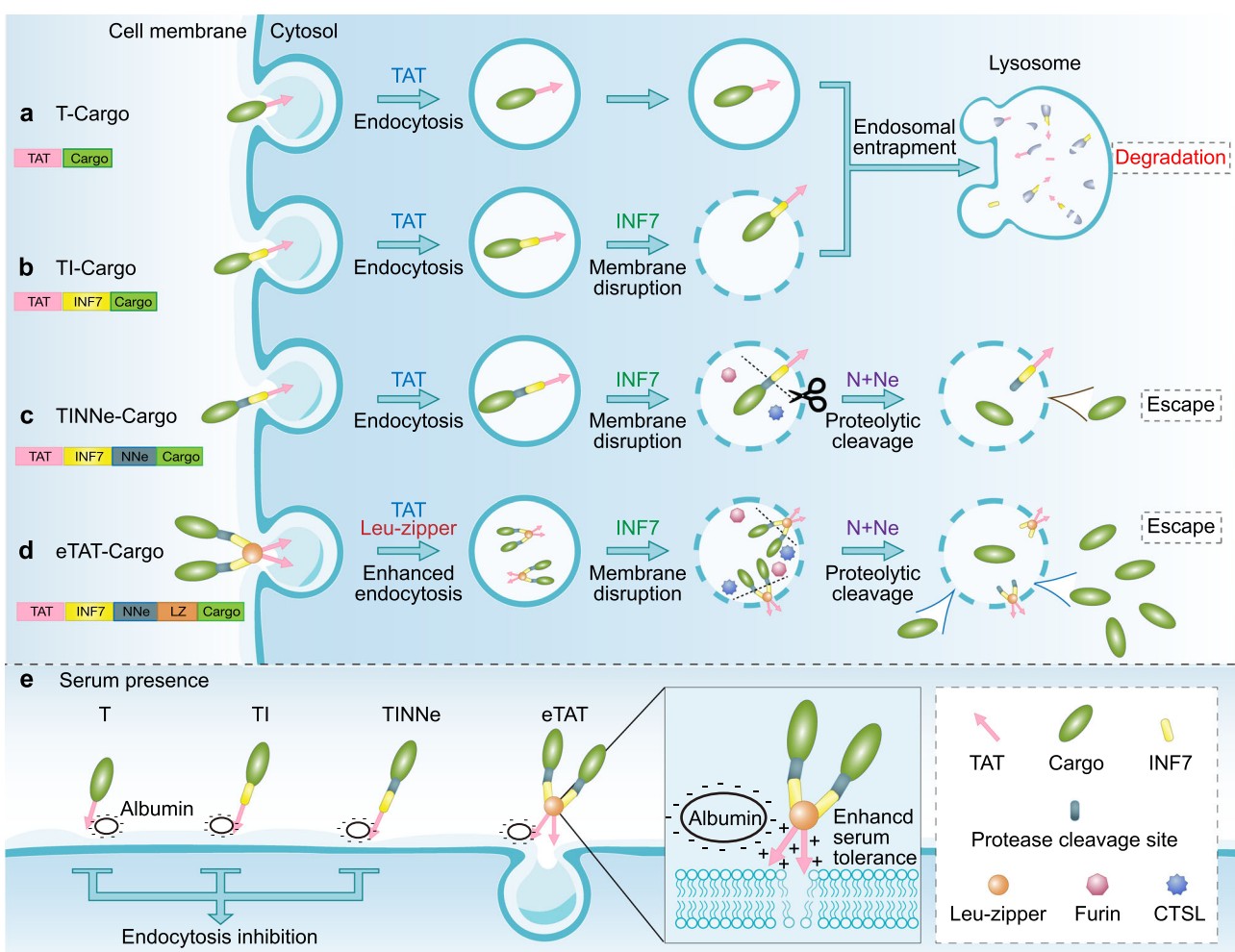

**Fig. 5 Schematic illustration of intracellular delivery mediated by eTAT and other TAT-based methods. a** The first step of intracellular delivery is the electrostatic interaction between TAT and membrane components on the cell surface (e.g. glycosaminoglycans), and then, cargos are internalised via TAT-induced endocytosis. However, T-cargos are entrapped in endosomes, and the majority of the cargo is ultimately targeted to lysosomes for degradation. **b** Even when the endosomal membrane is disrupted by INF7 (TI-Cargo), the majority of the cargo remains entrapped in endosomes because of the interaction between the endosomal membrane and TAT and/or that between the endosomal membrane and INF7. **c** When the proteolytic sites N and Ne are cleaved by endosome-localised proteases cathepsin and furin, the cargo is separated from TAT-INF7 and released rapidly from the disrupted endosome (TINNe-Cargo) into the cytosol. **d, e** The leucine zipper (LZ) can induce the formation of homodimers of the eTAT-cargo (TINNeL-cargo), which results in improved endocytosis (**d**) and weakened competitiveness of serum protein albumin (**e**) via increased local concentration of TAT that can interact with the cellular membrane components. CTSL: cathepsin L.

internalisation induced by the added functional modules, only the addition of PMAP (INF7) slightly increased the efficiency. Previous studies reported that charge of delivery sequence was critical for interaction between TAT and membrane components[50]; however, the fusion of INF7 to TAT sequence, which reduces the charge of delivery sequence, conversely resulted in the enhancement of intracellular uptake in the present study. The increases in cytosolic uptake by fusion of PMAP (HA2) to CPP has been also observed by Wölfl group and they proposed that the enhancement was due to conformational changes[51], which may enhance the membrane interaction or endosome disruption. Besides, the INF7 peptide could also act as a spacer or linker, which may make TAT more accessible for the interaction with the cell surface components (i.e. GAGs), and thus increase the intracellular uptake.

MCPPs was introduced into our chimaeric peptide strategy by using motifs of leucine zippers. As a homologous dimer was induced by addition of leucine zipper (Fig. 5d), eTAT exhibited not only highly efficient intracellular delivery but also high serum

tolerance (Fig. 2). Notably, the enhanced endocytosis conferred by dimerisation did not result in more cargo being released from endosomes, and endosomal entrapment restricted the delivery of protein to intracellular targets (Fig. 2d). Given this outcome, the eTAT system combines the benefits of enhanced internalisation due to higher localised concentration of positively charged CPPs and higher escape efficiency mediated by proteolytic cleavage, thus maximising the TAT-based delivery efficiency in vitro and in vivo. On the other hand, higher serum tolerance was also associated with the dimerisation of CPPs (Figs. 2e and 5e). MCPPs have been previously employed to increase cellular uptake; however, their impact on serum tolerance is not fully understood. In the current study, our results demonstrate that oligomerization of CPP-fused cargo was critical for serum tolerance (Fig. 2f). We speculate that MCPPs can increase the local concentration of a CPP where it interacts with cellular membrane components, leading to the alleviation of the competitiveness of albumin in serum. With high delivery efficiency and serum tolerance, eTAT shows great potential for use as a functional protein

delivery method both in vitro and in vivo. Ppm1b delivery by eTAT conferred nearly complete protection against TNF-induced necroptosis in L929 cells and against TNF-induced SIRS in BALB/c mice (Fig. 3). Furthermore, the clinical benefit of eTAT-Ppm1b was demonstrated here by the inhibition of necroptosis in acute acetaminophen-induced liver toxicity (Fig. 4). Intravenous administration of eTAT-Ppm1b twice, 2 h and 6 h after APAP challenge, significantly alleviated APAP-induced acute liver damage (challenge with a maximum nonlethal dose of APAP) and rescued ~60% of the mice challenged with the minimum absolute lethal dose of APAP (Fig. 4). In addition, eTAT-Ppm1b shows good safety. When healthy mice were intravenously administered eTAT-Ppm1b at tenfold the dose used to cure APAP-induced ALF, no obvious elevated serum ALT or AST level was observed (Supplementary Fig. 17).

The eTAT system described here presents a concept for the design of CPP-based agents that effectively deliver protein; however, it may require further optimisation, and there are still challenges to overcome. The eTAT system has remarkable features of flexibility, and any module can be substituted to meet user requirements. First, the TAT prototype consisting of CPPs can be replaced by newly discovered or invented CPPs that have greater delivery efficiency and lower cytotoxicity[52–54]. Second, other homo-oligomerizable peptides or small domains can be used to induce the formation of polymers with a higher degree of polymerisation than can be realised with a dimer[55–57], which lead to more effective endocytosis and higher serum tolerance. Third, the introduction of tissue- or cell-targeting peptides can enhance the capacity for delivering cargo to specific target tissues or specific types of cells for in vivo amplification[58–60]. In this study, we found that eTAT-Ppm1b injected intravenously was mainly distributed in the caecum, followed by the liver and kidney (Fig. 3). Ppm1b, but not eTAT or any modules within it, induced the caecal distribution of eTAT-Ppm1b (Supplementary Fig. 15); however, further experiments are needed to elucidate the reasons for the caecal accumulation of Ppm1b. We speculated that the choice of tissue- or cell-targeting peptides can alter the current biodistribution and that cargos delivered by eTAT can accumulate in the targeted tissues or cells. Finally, targeting a subcellular organelle by eTAT can be achieved via the introduction of organelle-targeting peptides, such as mitochondria-penetrating peptides[61] and an NLS domain. In this study, we demonstrate that cargos delivered into the cytosol by eTAT, such as GFP$_{1–10}$-NLS, were successfully translocated into the nucleus.

In summary, with significantly enhanced efficiency for intracellular delivery and enhanced serum tolerance, eTAT shows the potential to effectively deliver protein into the cytosol not only in vitro but also in vivo. eTAT and its improved/optimised versions will facilitate the development of biological macromolecular drugs for use against intracellular targets.

## Methods

**Antibodies and reagents**. Rabbit-anti-GFP (ab32146, dilution of 1:1000), anti-Rip3 (ab56164, dilution of 1:1000), anti-p-Rip3 (ab222320, dilution of 1:1000), and mouse anti-β-actin (ab6276, dilution of 1:1000) antibodies and goat anti-rabbit IgG H&L (HRP) (ab6721, dilution of 1:2000), anti-6 × His IgG (ab252883, dilution of 1:500) and goat anti-mouse IgG H&L (HRP) (ab205719, dilution of 1:2000) were obtained from Abcam. Mouse anti-GAPDH antibody (60004-1-Ig, dilution of 1:2000) was obtained from Proteintech. Donkey anti-mouse IgG H&L (Alexa Fluor 647) (A-31571, dilution of 1:500) was obtained from Thermo Fisher. Mouse TNF-α was purchased from Novoprotein, and zVAD was purchased from Selleck. Acetaminophen (APAP) and N-acetylcysteine amide (NAC) were obtained from MedChemExpress (MCE). Cell Counting Kit-8 (CCK-8) were obtained from Vazyme. Propidium iodide (PI), Hoechst 33342, and DAPI were purchased from Thermo Fisher. Gibson assembly master mix was purchased from NEB. X-tremeGENE was obtained from Roche.

**Plasmids**. GFP was amplified in pMSCV-loxp-dsRed-loxp-eGFP-Puro-WPRE plasmids (Addgene). Ppm1b-s (NM_011151) from mouse cDNA was amplified. Histone H3 from human cDNA was amplified. DNA encoding GFP$_{1–10}$ or mRuby3

was synthesised (Sangon Biotech). To determine the expression of the recombinant proteins, GFP$_{1–10}$-NLS-6 × His, GFP-6 × His, and Ppm1b-6 × His with different TAT-based delivery peptides fused to each N-terminus (or not) were cloned into a pET-21b (+) expression vector by the Gibson assembly method. For the preparation of lentivirus, mRuby3-histone H3-GFP β sheet 11 and Ppm1b were cloned into a pLV lentivirus vector without a tag by the Gibson assembly method.

**Expression and purification of recombinant proteins**. Plasmids derived from the pET-21b (+) vector were transformed into *Escherichia coli* BL21 (DE3) cells (New England Biolabs), and protein expression was induced with 1 mM IPTG administered at 25 °C for 15 h. The cells were harvested and resuspended in lysis buffer containing 20 mM phosphate buffer (pH 7.4), 150 mM NaCl, 5% glycerine and 1% Triton X-100. After cell lysis by ultrasonication and high-speed centrifugation at 25,000 × g for 20 min, the supernatant was filtered through a 0.45 μm microporous membrane. Subsequently, the His-tag-containing proteins were purified via Ni-NTA affinity chromatography (GE Healthcare), and the other proteins were purified via SP sepharose high performance chromatography (GE Healthcare). The concentration of purified protein was measured by Bradford assay, and aliquots were stored at −80 °C until use.

**Lentivirus production and infection**. To produce lentivirus, HEK-293T cells were transfected with pLV lentivirus vectors carrying the cDNA sequences of interest and lentivirus-packing plasmid (PMDL/REV/VSVG) using X-tremeGENE transfection reagent. After incubation for 12 h, the cell culture medium was changed to Dulbecco's modified Eagle's medium (DMEM) containing 10% foetal bovine serum (FBS) (Thermo Fisher). The virus-containing medium was collected 36 h later. For infection, 500 μL of virus-containing medium and 500 μL of fresh medium containing 10 μg/mL polybrene were added to cells plated in 12-well plates. The plates were centrifuged at 1500 g for 30 min and returned to the cell incubator.

**Cell lines and cell culture**. The HEK-293T human embryonic kidney cell line and L929 mouse fibrosarcoma cell line were generously provided by Professor Jiahuai Han (Xiamen University), which both obtained from ATCC (HEK-293T, CRL-3216; L929, CCL-1). The MA-104 embryonic rhesus monkey kidney cell line was obtained from ATCC (CRL2378.1). All cell lines were maintained in high-glucose DMEM supplemented with 10% FBS and 1% penicillin/streptomycin antibiotics (Thermo Fisher) at 37 °C in a humidified incubator with 5% CO$_2$.

To establish HEK-293T cells that stably express mRuby3-histone H3-GFP$_{11}$ in the nucleus (HEK-293T GFP$_{11}$), HEK-293T cells were infected with lentivirus carrying the mRuby3- histone H3-GFP$_{11}$ coding sequence. Twenty-four hours later, the medium was changed, and the cells were maintained and subjected to puromycin (2.5 μg/mL) selection for 48 h. Then, the drug-selected cells were analysed using flow cytometry (LSRFortessa X-20 Cell Analyser, BD). Individual cells presenting high and relatively uniform intensity of Ruby fluorescence were sorted into 96-well plates. Stable clones were selected, expanded and frozen.

**Animal care**. BALB/c mice were supplied by Shanghai SLAC Laboratory Animal Co., Ltd. The animals were housed in individual ventilated cages (IVCs) and maintained in a 12 h light-dark cycle, at a temperature of 22–25 °C and a relative humidity of 45–55%, and allowed free access to food and water. All animal experiments were carried out in compliance with the regulations of the Animal Welfare and Ethics Committee at Xiamen University (XMULAC20190152). Animal suffering was minimised or prevented at all times to improve their welfare.

**Endonuclear split GFP assay for evaluating endosomal escape**. HEK-293T GFP$_{11}$ cells were plated in 12-well plates at 5 × 10$^5$ cells per well and grown to 80% confluency in DMEM supplemented with 10% FBS. Then, the medium was discarded, and 1 mL of serum-free DMEM containing GFP$_{1–10}$-NLS-related recombinant proteins at the indicated concentrations was added. After treatment for 3 h, the recombinant protein solution was aspirated, and the cells were washed 3 times with DMEM containing 10 U/mL heparin. Subsequently, the cells were cultured in DMEM supplemented with 10% FBS at 37 °C for 9 h. Then, the cells were detached and analysed using a flow cytometer (LSRFortessa X-20 Cell Analyser, BD). The cells displaying a normal morphology were gated first; these cells expressing mRuby3 were gated, and their GFP fluorescence was analysed. The calculated mean fluorescent intensity (MFI) and the percentage of green fluorescence-positive cells were recorded. Gating strategy to sort GFP-positive cells in HEK-293T-GFP$_{11}$ treated with GFP$_{1–10}$-NLS-related recombinant proteins was presented in Supplementary Information (Supplementary Fig. 19a). The data were collected by using BD FACSDiva software, and analysed by using FlowJo software.

When collected for immunoblot analysis at the indicated time points during the treatment procedure, the cells were washed three times with DMEM containing 10 U/mL heparin before detachment. When an endonuclear split-GFP assay was performed to test the impact of the serum on intracellular delivery, and the cells were treated with GFP$_{1–10}$-NLS-related recombinant proteins diluted in DMEM and supplemented with FBS at the indicated percentage.

**Protein transduction of GFP and Ppm1b**. The indicated cells were grown to 80% confluency in 12-well plates, and the medium was removed. Then, the cells were treated with 1 mL of GFP-related or Ppm1b-related recombinant proteins at the given concentration diluted in serum-free DMEM or DMEM supplemented with FBS at the indicated percentage for 3 h at 37 °C. After treatment, the medium containing GFP-carrying or Ppm1b-carrying proteins was aspirated, and the cells were washed three times with DMEM containing 10 U/mL heparin. For GFP analysis, the cells were detached, and the calculated mean fluorescent intensity (MFI) and the percentage of green fluorescence-positive cells were recorded using a flow cytometer (LSRFortessa X-20 Cell Analyzer, BD). Gating strategy to sort GFP-positive cells in HEK-293T treated with GFP-related recombinant proteins was presented in Supplementary Information (Supplementary Fig. 19b). The data were collected by using BD FACSDiva software, and analysed by using FlowJo software. For Ppm1b analysis, the cells were detached, and the level of Ppm1b was determined by WB using anti-6 × His antibody.

**Immunoblot analysis**. The same number of cells ($5 \times 10^5$ cells per well in 12-well plates) were seeded in cell culture plates. After treatment with the conditions specified in the figure legends, all the cells in a plate were detached and resuspended in PBS. Then, the cell numbers were determined and adjusted to $1 \times 10^5$ cells per 100 μL of lysis buffer (50 mM Tris-HCl, pH 7.4; 150 mM NaCl; 1% Triton X-100; 0.1% SDS; 1 mM EDTA, 2.5 mM sodium pyrophosphate; 1 mM β-glycerophosphate; and 1 mM Na$_3$VO$_4$). After incubation on ice for 30 min, the cell lysate was centrifuged at $15,000 \times g$ for 15 min at 4 °C, and an equal volume of supernatant was loaded into each lane and then used for standard SDS-PAGE. The proteins were blotted onto PVDF membranes (Millipore) and probed with primary antibody specific to the target protein as indicated. Bound antibody was detected using anti-rabbit IgG or anti-mouse IgG conjugated to horseradish peroxidase and visualised by an ImageQuant LAS 4000 scanner (GE-Healthcare) with enhanced chemiluminescence (Pierce ECL western blotting substrate, Thermo Fisher).

**High-performance size exclusion chromatography (HPSEC)**. The purified proteins and protein molecular weight standards (GE Healthcare, Little Chalfont, UK) were analysed using an Agilent 1200 HPLC system (Agilent Technologies, Santa Clara, CA) with an analytical TSK Gel G3000PWXL (TOSOH, Tokyo, Japan) to analyse the molecular weight. In the system, the elution buffer was PBS + 0.01% SDS or 0.4 M NaCl with a flow rate maintained at 0.5 mL/min, and the absorbance at 280 nm was monitored for the detection of protein in the eluent.

**Cytotoxicity assay**. Cell viability was measured using cell counting kit-8 (CCK-8). The kit consists of a water-soluble tetrazolium salt (tetrazolium-8) that is used to quantify the number of live cells by producing a soluble yellow formazan dye upon bio-reduction in the presence of an electron carrier. HEK-293T cells were seeded into 96-well plates at a density of $5 \times 10^3$ cells per well and exposed to 1, 2, 5, 10, 20 and 50 μM of different TAT-containing recombinant proteins diluted in serum-free DMEM for 24 h. Subsequently, the cells were washed with PBS and further incubated with CCK-8 solution for 2 h. The optical density values were then measured at 450 nm using a microplate reader (Thermo Fisher). The results are presented as percentages of the control value obtained on the basis of the absorbance of untreated cells.

**Cell death assay**. Cell death was analysed for loss of membrane integrity by measuring PI uptake. To measure TNF-induced cell death, L929 cells treated with or without Ppm1b were stimulated with TNZ in complete DMEM (10 ng of mouse TNFα and 20 nmol zVAD per mL of medium) for 5 h, trypsinized, collected by centrifugation, and resuspended in serum-free DMEM containing 5 μg/mL PI. After incubation for 15 min at room temperature, the level of PI incorporation was analysed by a Beckman-Coulter CyAn ADP flow cytometer. PI-positive cells were considered as dead cells. Gating strategy to sort PI-positive cells in L929 treated with Ppm1b-related recombinant proteins was presented in Supplementary Information (Supplementary Fig. 19c). The data were collected and analysed by using Summit v4.3 software.

To test the effect of Ppm1b protein transduction on TNF-induced cell death, L929 cells were plated and incubated overnight, and then, the cells were treated with 5 μM Ppm1b-related recombinant proteins in serum-free DMEM for 3 h. Subsequently, the treated cells were washed three times with DMEM containing 10 U/mL heparin and stimulated with TNZ. For this test, Ppm1b-overexpressing L929 cells were used as positive controls. For the overexpression of Ppm1b, L929 cells were infected with lentivirus carrying the Ppm1b coding sequence for 48 h and then replated 12 h before TNZ stimulation to remove spontaneously dying cells.

**Quantitative determination of proteins**. After treatment, cells were counted ($1 \times 10^5$ cells), harvested and lysed in 100 μL of lysis buffer (50 mM Tris, pH 7.5; 2 mM EDTA; 2 mM DTT; and 0.1% Triton X-100). The lysate was transferred to a 96-well plate, and the fluorescence emission intensity was measured ($\lambda_{exc} = 488$ nm/$\lambda_{em} = 530$ nm) using a plate reader (Paradigm Detection Platform). The amount of intracellular protein in the lysate was identified based on corresponding standard curve. Finally, the intracellular protein concentration was

calculated using the following Eq. (1):

$$\text{Intracellular protein concentration} = \frac{\text{Absolute amout of protein in cell lysate}}{\text{Cell count in lysate} \times \text{Cell volume}} \quad (1)$$

**TNF-induced SIRS mouse model**. Six- to eight-week-old female mice (average weight of ~20 g) were injected intravenously with 15 μg of mouse TNF-α diluted in endotoxin-free PBS and then continuously observed. Mouse survival was specifically determined every hour. For Ppm1b pretreatment, the mice were injected with 5 nmol Ppm1b-related recombinant proteins diluted in 250 μL of endotoxin-free PBS or the same volume of PBS as a control via the tail vein 2 h before SIRS induction. Mock mice were not treated with any reagents. The investigator was blinded to the mouse allocation when pretreating the mice with Ppm1b, injecting them with TNF and counting the number of mice that died.

**Histology**. The mice were euthanized 12 h post-TNF treatment. After euthanasia, the caecal tissues were collected immediately and fixed in 4% paraformaldehyde for 24 h. The fixed samples were embedded in paraffin and cut into 4 μm sections. The sections were sliced, mounted on microscope slides, and then stained with haematoxylin and eosin (H&E). The images were captured with a Leica FDM2500 microscope. Then, the degree of caecum damage was assessed by a method described in previous study[62]. In briefly, caecal damage is featured by epithelial cell death in lumen, regenerative changes in surface epithelium, loss of mucus layer and goblet cells, and edema in submucosa. Taking into account all histological characteristics, a damage score ranging from 0 (normal) to 3 or 4 (abnormal) in these four parameters was given to each mouse. The sum of average scores of each sample as evaluated by four investigators was calculated as the final caecum damage score for one sample. The investigators were blinded to the allocation when performing the histology experiments and scoring the tissue damage.

**Ex vivo biodistribution imaging**. Ppm1b, T-Ppm1b, eTAT-Ppm1b and eTAT-GFP were labelled with the fluorescent dye Cy5 using Cy5 NHS ester (Lumiprobe) according to the manufacturer's instructions. BALB/c mice were intravenously injected with 20 nmol Cy5-labelled proteins in 500 μL of endotoxin-free PBS and then euthanized 24 h later. Immediately after euthanasia, the organs were collected, and quantitative imaging was performed using an IVIS spectrum system (PerkinElmer).

**Immunohistochemistry**. Female BALB/C mice were intravenously administered 20 nmol recombinant protein. Five hours after injection, the liver and caecum were harvested, washed with PBS and fixed in 4% paraformaldehyde for 24 h at 4 °C. The paraffin sections were prepared by the methods described above (TNF-induced SIRS mouse model). Before antibody incubation, antigens were retrieved. The slides were dipped in PBS for 10 min and permeabilized using 0.3% Triton X-100 in PBS for 6 min, and blocked for 1 h (10% normal goat serum in PBS containing 0.05% Triton X-100). The sections were then incubated with anti-6 × His IgG diluted 1:500 in antibody cocktail solution (50% blocking solution, 0.05% Triton X-100 in PBS). Following primary antibody incubations, the slides were washed three times with washing buffer (1% blocking solution in PBS containing 0.05% Triton X-100), 10 min each, and incubated with the Alexa 647-conjugated secondary antibodies diluted 1:500 in antibody cocktail solution for 1 h at room temperature protected from light. Nuclei were stained with DAPI for 3–5 min following antibody binding. Images were acquired using an Invitrogen EVOS M7000 fluorescence microscope (Thermo Fisher).

**APAP-induced acute liver failure model**. Five- to seven-week-old BALB/c mice (average weight of ~18 g) were used as APAP-induced acute liver damage models. APAP was dissolved in 10% DMSO at a concentration of 400 mg/ml. Then, APAP was diluted in endotoxin-free PBS to the indicated concentrations and warmed to 37 °C before injection. To determine the maximum nonlethal dose and the absolute lethal dose, overnight-fasted mice were injected intraperitoneally with different doses, from 300 to 900 mg APAP per kg of body weight. The mortality of the mice was monitored after the APAP injection, and survival was determined every hour.

To therapeutically treat APAP-induced acute liver damage, overnight-fasted mice were intraperitoneally injected with 500 mg of APAP per kg of body weight, the maximum nonlethal dose determined in this study, and then treated with 5 nmol Ppm1b-related recombinant proteins in 250 μL of endotoxin-free PBS twice, 2 h and 6 h post-APAP injection. Twelve hours after the APAP injection, the mice were euthanized, and blood was collected to measure serum ALT and AST levels. As a control group, mice injected with APAP were treated with PBS instead of Ppm1b, and these mice were injected with 0.9% DMSO in PBS, the concentration of DMSO in the APAP solution used for injection, followed by PBS treatment. The mock mice were not treated with any agents. The investigator was blinded to the allocation when measuring the aminotransferase levels.

To administer therapy to prevent APAP-induced mouse death, overnight-fasted mice were intraperitoneally injected with 800 mg APAP per kg of body weight, the absolute minimum lethal dose determined in this study, and then treated with 5 nmol eTAT-Ppm1b in 250 μL of endotoxin-free PBS twice, at 2 h and 6 h post-APAP injection. The control mice were injected with APAP and treated with PBS

or 600 mg NAC per kg of body weight instead of eTAT-Ppm1b. The mortality of the mice was monitored, and survival was determined every hour. The investigator was blinded to the allocation when treating the mice with eTAT-Ppm1b, NAC or PBS and when counting mouse deaths.

**Statistical analysis**. GraphPad Prism software version 6.0 and 7.0 (GraphPad Software) were employed for the statistical analyses. The data are presented as the means ± s.e.m. Unpaired two-tailed Student's *t* tests were used to compare the differences between two groups. Survival curves were plotted using the Kaplan–Meier method, and significance was calculated by log-rank (Mantel-Cox) test. Differences in the compared groups were considered significantly different when the $P < 0.05$. *, **, *** and **** indicate $P$ values less than 0.05, 0.01, 0.001 and 0.0001, respectively.

**Reporting summary**. Further information on research design is available in the Nature Research Reporting Summary linked to this article.

## Data availability

All data supporting this study are available in the main figures and the Supplementary Information files or from the corresponding author upon reasonable request. Source data are provided with this paper in a single excel file. A reporting summary for this article is available as an Additional Information file. Source data are provided with this paper.

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

## Acknowledgements

We thank Professor Jiahuai Han (Xiamen University) for kindly providing the L929 mouse fibroblast line and the HEK-293T human embryonic kidney cell line. This research was supported by the Ministry of Science and Technology of China (2020YFC1316800), the National Natural Science Foundation of China (31971369), the National Key Programme for Infectious Disease of China (2018ZX10732401-003-008), and the Natural Science Foundation of Fujian Province of China (2019J02004).

## Author contributions

X.N.S., G.S.X. and Y.Q. conceived the study and designed the experiments; Z.J., Z.T.Y., Z.S.Y., T.R. and C.Y.X. provided intellectual advice on the project and experimental design; Y.L.Q. performed preliminary works; Y.S.Y., P.H.F. and J.J.L. designed, purified recombinant proteins under the supervision of L.T.D., Y.S.Y.; Y.H. performed cell culture experiments under the supervision of L.T.D.; Y.S.Y., Z.Y.L. and W.S.J. performed the flow cytometry experiments; Y.S.Y. and Y.H. performed the immunoblotting experiments; Y.H., P.H.F., R.S.L., L.G.X. and C.B.B. performed the in vivo transduction experiments; L.T.D. supervised and analysed data; Y.H. drafted the paper, which was edited and revised with G.S.X. and L.T.D.; all authors discussed and refined the paper.

## Competing interests

The authors declare no competing interests.
