## [Peer Review File · Nature Communications]

Reviewers' Comments:

Reviewer #1:

Remarks to the Author:

These authors highlight the reasons protein delivery to cells cannot easily be accomplished, even by CPPs. Endosomal entrapment (poor release from endosomes) and loss of CPP delivery activity in serum. They have developed a hybrid approach using a CPP, pH sensitive membrane active peptide combined with endosome-specific protease sites and a leucine zipper for dimerization to create a protein delivery vehicle that appears to deliver proteins to cell cytosol in vitro and in vivo. This is a very interesting study, and there are multiple insights that will be of broad interest to researchers in protein delivery. However, there are some issues the authors will need to address to strengthen the paper before publication.

1) The most significant issue of the paper is shown in Supplemental Figure 1. These data show very clearly that better delivery does not mean that individual cells get more protein delivered to them. All constructs tested, including the very poor ones, deliver a similar amount of cargo to individual cells. The better constructs simply deliver this amount of cargo to more cells. This fact is not addressed in the main paper, yet it holds important clues to something very fundamental about protein delivery that we do not understand. Many researchers have observed the same phenomenon. Delivery is stochastic, and cooperative at the cell level, not at the individual endosome level. Improving protein delivery to cells is not about improving the efficiency of release from individual endosomes. It is not about delivery of more protein to individual cells. With respect to cargo delivery, cells act cooperatively. Improving delivery is about delivering a bolus of protein to a greater fraction of cells. Yet, this entire paper (and most other papers on cargo delivery) presents the problem and discusses the results it as if the modifications carried out incrementally improve the amount of protein delivered to each cell at the level of individual endosomes. This is not correct. The fundamentally important observation in Supplemental Fig 1 must be a central part of the whole paper. Introduction, results, and discussion.

2) The fluorescence microscopy images shown for delivery of full length GFP seem not to show the same effect as described in point 1 above. Is this true? The authors should explain this.

3) One serious problem with almost all protein delivery assays is a lack of absolute measures of delivery. For example, Figures 1c and 1d show mean fluorescence intensity as a ratio of intensity observed with various constructs, to that observed with a construct that delivers little to no GFP1-10. Relative MFI is the ratio of a measurement over a small number. The reader of this paper needs to know what MFI=10 means in terms of absolute amounts. One could lyse cells, and use absolute fluorescence of the lysate compared to standard solutions of known concentrations of GFP to determine the absolute amount of GFP1-10 delivered. But given the stochastic nature of delivery (Supplemental Figure 1) a microscopic technique might be better suited. This problem of quantitation is even worse for enzymes because their effect is greatly amplified.

4) The writing syntax and sentence structure is very poor throughout the manuscript. The revised paper will need significant editing and polishing of the English. To provide one example, the opening paragraph of the discussion has several sentences that are nonsensical at present.

4) The paper contains a number of statements that are essentially unsupported speculation. I feel that they should be removed if they cannot be supported by some evidence from the literature or from these experiments. Such speculations include:

Line 119-120 "The CPP dimer might engage with the cellular membrane through multivalent interactions, but with, serum albumin in a monovalent manner; thus, the competitiveness of the serum is alleviated."

Lines 253-256 "Unexpectedly, endosomal escape efficiency of this dimer is quite low than observed at co-incubation approach of dTAT30. The reason for this is unclear, it is possible that the effect of cargo molecules on the interaction between TAT with BMP (bis(monoacylglycero)phosphate)."

5) The animal experiments are very impressive and add a lot to this work. Again, we lack any quantitative measure of delivery. Is there a way to determine how much enzyme must be

delivered to prevent the acute damage that drives the observed phenomenon?

6) Can the authors use the Cy5 fluorescence to determine the amount of enzyme delivered to cells? Can fluorescence microscopy be done on tissue sections to directly demonstrate and quantify cargo delivery to cells in vivo.

Reviewer #2:

Remarks to the Author:

Yu et al. report on a novel CPP(TAT)-based intracellular protein delivery system (eTAT) consisting of a CPP, PMAP, endosome-specific protease sites and a leucine zipper, showing enhanced cytosolic protein, phosphatase 1B (Pmp1b), delivery efficiency and serum tolerance in vitro and in vivo without toxicity, suppressing the TNF- α -induced systemic inflammatory response and acetaminophen-induced acute liver failure in the mouse model.

They provide a strategy to facilitate the development of CPP-based delivery platform on proteins for intracellular targets. The work is well performed and timely.

However, my concern is that it is difficult to see clear advantage of this method as compared to previously published protein delivery systems, e.g. TAT peptide and multiple others. I suggest to carry out additional studies in order to confirm such advantage.

Reviewer #3:

Remarks to the Author:

Authors have designed and refined Tat-peptide-based vector for delivering into cells protein cargoes covalently coupled to peptide. First, the widely used CPP was supplemented with PMAP sequence, and two protease cleaving sites for facilitating release of cargo proteins from endosomal entrapment. Later the dimerisation-inducing motif was included in peptide vector to increase the uptake by cells and reduce sensitivity towards serum albumin that is known to inhibit Tat-mediated delivery into cells.

The efficiency of elaborated eTAT vector was verified by delivering GFP and its split version into model cell-lines. Finally, eTAT was used for delivery of Pmp1b to suppress TNF-induced necroptosis in vitro and in vivo. Moreover, eTAT-Pmp1b also increased survival of mice in acetaminophen-induced acute liver failure model.

Major comments:

1. Authors claim that the used strategies for improving Tat-peptide mediated delivery are novel. However, higher delivery efficacy of multivalent CPPs compared to monovalent peptide has been earlier demonstrated by several laboratories e.g. of Futaki and Gariépy. Their works have to be referred to in introduction. The idea of using cleavage by protease for activation of CPP construct has been repeatedly exploited in the field of CPPs, and was pioneered by R. Tsien group, who perhaps also deserves mentioning.
2. Fig. 1a, a scheme of recombinant cell-penetrating protein, does not reveal where the polyhistidine tag that is used for purification of protein, is placed in the sequence. Please add! All data in manuscript suggest that histidine tag is still present on all used protein constructs, and it is not removed. Histidine residues are known to serve as a proton sponge and facilitate escape from endosomes, was its impact considered?
3. Fig 1h. The intensity of GFP signal is substantially higher for TI-GFP compared to T-GFP after incubation MA104 cells with the respective protein constructs. The efficiency of cationic CPPs is dependent on the charge of peptide (R and K in sequence). Addition of INF7 to TAT sequence reduces the charge of targeting sequence from 8 to 4, which typically strongly impairs internalisation of peptide/protein construct. Here the fluorescence microscopy shows opposite – protein carrying the transducing sequence with lower positive charge is taken up by cells more efficiently than one with higher charge. Please explain this discrepancy in discussion part. Please show more than 1 or 2 cells in new version of Fig 1h.
4. Line 123 ... As analyzed by SDS-PAGE, T-GFP and TL-GFP are monomers and dimers

respectively (Fig. 2b).

It is highly improbable that in SDS-PAGE the latter protein construct is in the form of dimer only. The Supplementary Figures 5, 6 and 8 show quite opposite.

5. Line 191 ... Twenty-four hours after administration, eTAT-Ppm1b was mostly distributed in the caecum ...

This is rather surprising observation. Has that been shown for other CPPs or CPP-protein constructs earlier? Please comment on this in discussion.

6. Line 234 ... the endocytosis of the cargo was mediated by the interaction between TAT and the cell membrane (Fig. 5a)

Does TAT peptide/motif interact directly with membrane? It has been repeatedly shown that proteins are involved in interaction and penetration of this CPP across membranes

7. Discussion contains multiple factual errors and difficult-to-understand sequences. It requires revision with focussing on obtained results and leaving out ample putative developments of presented protein delivery vector.

In multiple places the claims of authors are presented without the respective reference/citation (e.g. line 44-45, 50 etc.).

The manuscript contains many typos (e.g. line 33), citation errors (e.g. line 59, line 118 [18]) and sentences that are difficult to read (e.g. lines 24-28, 37-40, 250-254) or are left unfinished (e.g. lines 255-256).

Specific comments:

1. Line 52-53 ... After endocytosis mediated by CPP, the majority (approximately 90%) of the cargos are entrapped inside endosomes.

Does it mean that 10% liberates and is active? Please provide reference for 90%.

In general, one or two logs lower escape/activity is considered realistic.

2. Line 70-72 ... Therefore, we believe the eTAT system or its improved version composed of multifunctional chimeric peptide has potential to become a major enabling strategy for protein delivery in biological research and therapeutic application.

Exaggeration, considering how large quantities of protein construct (per kg) were required to achieve effect in mouse model.

3. In Figure 1e type of actin?

The level of actin varies substantially between lines. Please specify (in Materials and Methods) how the amount of protein that was applied to gel was equalised.

4. Figure 1f. Fluorescence of GFP in split assay peaks at 9 h time-point and starts to decrease from there, which occurs surprisingly quickly. Please comment this in discussion.

5. Lines 118-119 ... homodimerization capacity was employed to mediate the dimerization of the CPP-fused proteins.

This approach has been used with CPPs earlier. Please add reference.

6. Lines 119-121 ... The CPP dimer might engage with the cellular membrane through multivalent interactions, but with, serum albumin in a monovalent manner; thus, the competitiveness of the serum is alleviated.

What gives ground for this assumption? Has it been shown experimentally?

For verifying this assumption, the authors are advised to analyse the elution of dimeric protein construct (after incubation with serum or not) from SEC column. Comparison of that with elution profiles of monomeric from should reveal, whether both proteins associate only one molecule of albumin.

7. Line 170 ... eTAT showed the highest efficiency for Ppm1b delivery (Fig. 3a and supplementary fig. 9).

Actually neither figure shows delivery efficiency.

8. line 210 ... and were intravenously injected with 20 nmol Ppm1b protein twice
Material and Methods says that 5 nmoles (line 451). Which dose was actually used?

Authors' responses to reviewers' comments for the manuscript:

**Efficient intracellular delivery of proteins by a multifunctional
*chimaeric peptide both in vitro and in vivo***

Siyuan Yu¹, Han Yang¹, Tingdong Li¹, Haifeng Pan¹, Shuling Ren, Guoxing Luo, Jinlu Jiang, Linqi Yu, Binbing Chen, Yali Zhang, Shaojuan Wang, Rui Tian, Tianying Zhang, Shiyin Zhang, Yixin Chen, Quan Yuan*, Shengxiang Ge*, Jun Zhang, and Ningshao Xia*

Our revisions are described below in a point-by-point manner. For easier reading, the reviewers' comments are marked **in black**, responses are marked **in blue**, figure legend are marked **in red**. Questions are copy-pasted from the *Nature Communications* Editor's decision correspondence. Changes in the main text of the revised MS are highlighted **in yellow**.

Reviewer #1 (Remarks to the Author):

These authors highlight the reasons protein delivery to cells cannot easily be accomplished, even by CPPs. Endosomal entrapment (poor release from endosomes) and loss of CPP delivery activity in serum. They have developed a hybrid approach using a CPP, pH sensitive membrane active peptide combined with endosome-specific protease sites and a leucine zipper for dimerization to create a protein delivery vehicle that appears to deliver proteins to cell cytosol *in vitro* and *in vivo*.

This is a very interesting study, and there are multiple insights that will be of broad interest to researchers in protein delivery. However, there are some issues the authors will need to address to strengthen the paper before publication.

Re: Thanks very much for your appreciation on the value of our work. We have carefully revised the manuscript according to the comments from you and the other two reviewers. Point-by-point responses to the comments are listed below.

1) The most significant issue of the paper is shown in **Supplemental Figure 1**. These data show very clearly that better delivery does not mean that individual cells get more protein delivered to them. All constructs tested, including the very poor ones, deliver a similar amount of cargo to individual cells. **The better constructs simply deliver this amount of cargo to more cells.** This fact is not addressed in the main paper, yet it holds important clues to something very fundamental about protein delivery that we do not understand. Many researchers have observed the same phenomenon. Delivery is stochastic, and cooperative at the cell level, not at the individual endosome level. **Improving protein delivery to cells is not about improving the efficiency of release from individual endosomes. It is not about delivery of more protein to individual cells.** With respect to cargo delivery, cells act cooperatively. Improving delivery is about delivering a bolus of protein to a greater fraction of cells. Yet, this entire paper (and most other papers on cargo delivery) presents the problem and discusses the results it as if **the modifications carried out incrementally improve the amount of protein delivered to each cell at the level of individual endosomes. This is not correct.** The fundamentally important observation in Supplemental Fig 1 must be a central part of the whole paper. Introduction, results, and discussion.

2) **The fluorescence microscopy images shown for delivery of full length GFP seem not to show the same effect as described in point 1 above.** Is this true? The authors should explain this.

Re: We sincerely appreciate your valuable comments, because your comment 1-2 touch the same issue, thus we sought to address them together as follow:

I. We claimed that better construct increased the amount of protein delivered into each cell, however, as show in supplementary figure.1, it seemly only delivered this amount of cargo to more cells.

Firstly, the FACS analysis in original Fig.1e demonstrated that our modification, such as introduction of two cleavage sites (TINNe-GFP₁₋₁₀-NLS), increased mean fluorescence intensity (MFI) of total treated cells, indicating that more protein has been delivered into cells. This observation was shown more clearly by the shifting of peak in the histograms,

which suggest that more proteins were delivered into each cell (**Please see Fig.R1a below**). Thus, we reasoned that the enhancement of MFI analyzed by FACS could support the claim that better construct increased amount of protein delivered to each cell. In the previous research by Wimley Group (Kauffman et al. (2018) Nat Commun, 9: 2568.), the level of MFI of total treated cells was to evaluate the amount of cargo delivered to each cell as well.

Nevertheless, as you correctly pointed out, the original supplemental figure.1 showed that enhanced MFI of cell population would be accompanied by increase of the percentage of GFP-positive cells. However, when in delivery of full length GFP, all the cells seemly exhibit GFP fluorescence with different intensity between various constructs (**comment 2**). We speculated that it was attributed to two potential drawbacks of the endonuclear split GFP assay in our original submission: **lower fluorescence intensity of GFP after complementation and undetermined level of GFP₁₁ in the nucleus.** **1)** Due to lower fluorescence intensity of GFP after complementation, only when enough escaping GFP₁₋₁₀-NLS entered into nucleus, the level of GFP could reached its detection threshold, in other words, green fluorescence in the nucleus could be imaged by microscope or identified as the GFP positive by FACS. As shown in Fig.R1a and 1b, enhanced MFI within total cells mediated by higher concentration of TINNe-GFP₁₋₁₀-NLS was accompanied by increase of percentage of GFP-positive cells. Similar phenomenon was observed in other study using Split-GFP assay as well (Lonn et al. (2016) Sci Rep, 6: 32301.). **2)** On the other hand, due to absence of any indicator for expression of GFP₁₁, even though original HEK-293T-GFP₁₁ cell line was identified by FACS after transfection of GFP₁₋₁₀-NLS-expressing plasmid previously, we speculated that the level of GFP₁₁ was still lower (even not express) in the majority of HEK-293T-GFP₁₁ cells. This problem probably caused that only a fraction of cells shows GFP fluorescence after treatment with GFP₁₋₁₀-related proteins as well, although enough GFP₁₋₁₀-NLS entered into nucleus.

To overcome the second drawback of original endonuclear split GFP assay in the original MS, we constructed a new HEK-293T GFP₁₁ cell line, in which full length mRuby3 (red fluorescence) was introduced in the N terminal of Histone H3 and GFP₁₁, thus the signals of Ruby could be used to determine the expression of GFP₁₁, and detect the nucleus (Please see Fig.R1c, R1d below or revised Fig. 1b, new supplementary Fig. 1). Moreover, HEK-293T cell line stably expressing higher and more uniform intensity of mRuby3 was selected using FACS (Please see Fig.R1e below or new supplementary figure.1). Finally, we investigated the intensity of GFP in the updated assay, as shown in Fig.R1b below, novel HEK-293T-GFP₁₁ shows higher MFI than that of original assay, when incubated with same concentration of TINNe-GFP₁₋₁₀-NLS. In addition, FACS analysis (Please see Fig.R1f below) also show that almost all the novel HEK-293T-GFP₁₁ cells exhibit green florescence in both concentration groups, although the MFI was lower when the cells were treated with 1 μ M TINNe-GFP₁₋₁₀-NLS. **Collectively, we reason that our updated assay could avoid this confused observation as much as possible.**

Given that results involved original Split-GFP assay caused confusion above, in this revision, we re-performed all relevant experiments based on our novel endonuclear Split-GFP assay (i.e., Fig.1b, 1c, 1d, 1f, 2d, 2i, 2j and 2k; Supplementary Fig.1, 3, 4, and 5). **Consistent with the results in original MS, proteolytic removal of CPP-PMAPs in**

endosomes indeed increase the MFI of HEK-293T-GFP₁₁ as well. However, differ from the original data, when introduction of INF7 or more modules (TI-, TIN-, TINe-, TINNe-), almost all the HEK-293T-GFP₁₁ exhibit observed green fluorescence, and the TINNe-GFP₁₋₁₀-treated cells showed highest MFI (Please see below or revised Fig.1d). Now we have replaced all the relevant figures in original MS with new set of data using novel assay, in particular revised supplementary Fig. 3 below (former supplementary Fig.1). The results in revised supplementary Fig. 3 also show that more proteins were delivered into cells with the addition of PMAP and proteolytic sites.

With respect to the discrepancy between the delivery of Split-GFP and full-length GFP, as explained above, it was mainly attributed to lower fluorescence intensity of GFP after complementation, while the intensity of full length GFP is strong enough to be observed in the almost all the treated cells by using microscope. We reason this concern could also be addressed after the introduction of updated Split-GFP assay (Please see below or revised Fig.1h).

Fig. R1 The construction of novel Split-GFP assay. **a** The percentage of green fluorescence-positive cells within HEK-293T GFP₁₁ treated with 1 μM or 5 μM TINNe-GFP₁₋₁₀-NLS in the original Split-GFP assay. **b** The MFI within total cells treated with 1 μM or 5 μM TINNe-GFP₁₋₁₀-NLS in the original or updated Split-GFP assay. **c** Schematics of the updated endonuclear Split GFP assay. Only after GFP₁₋₁₀-NLS was released from the endosome and translocated into the nucleus of HEK-293T cells with stable expression of histone H3-GFP β-sheet 11 (HEK-293T GFP₁₁), the fluorescence of the complete GFP in the nucleus was observed and co-localized with the fluorescence of mRuby3. **d** Representative fluorescence microscopy images of novel HEK-293T GFP₁₁ cell line. Nucleus was stained with hoechst. **e** The percentage of mRuby3-positive cells of HEK-293T-GFP₁₁ (organ) or wild type HEK-293T (grey) analyzed by FACS. **f** The percentage of green fluorescence-positive cells within HEK-293T GFP₁₁ treated with 1 μM or 5 μM TINNe-GFP₁₋₁₀-NLS in the updated Split-GFP assay. The results shown in **b** are the means ± s.e.m.; *** *P* < 0.001, **** *P* < 0.0001.

Revised Fig. 1 Proteolytic removal of CPP-PMAP in the endosomes enhances the escape efficiency of protein cargos. d MFI (left panel) within total HEK-293T cells and corresponding percentage of green fluorescence positive-cells (right panel) after the treatment GFP₁₋₁₀-NLS-related proteins containing two cleavage sites. GFP₁₋₁₀-NLS-related proteins were used at a concentration of 5 μ M. The results shown in **d** are the means \pm s.e.m: *** $P < 0.001$.

Revised Supplementary figure. 3 Representative fluorescence microscopy images of the treated HEK-293T GFP₁₁ in Figure. 2d. The HEK-293T GFP₁₁ were incubated with GFP₁₋₁₀-NLS-related recombinant proteins (5 μ M) for 3 h then washed, and imaged using fluorescence microscope at 12 h post incubation onset.

Revised Fig. 1 Proteolytic removal of CPP-PMAP in the endosomes enhances the escape efficiency of protein cargos. h. MFI (left panel) within total HEK-293T cells and corresponding

percentage of green fluorescence positive-cells (right panel) after the treatment of different GFP-related proteins(n=3). Proteins were used at a concentration of 1 μ M. Results in h shown are means \pm s.e.m: * $P<0.05$ and NS, no significant difference.

II. We claimed that that modifications performed incrementally improve the amount of protein delivered to each cell at the level of individual endosomes. However, delivery is stochastic, and cooperative at the cell level, not at the individual endosome level.

With respect to this concern, please allow us to explain that the process of cargo delivery by CPPs could be divided into two steps generally: internalization (endosome formation) and subsequent endosomal escape (cytosolic delivery and nucleus localization). Hence, the final delivery efficacy depended on the efficiency of these two steps. For example, in the Split-GFP assay, the MFI of HEK-293T-GFP₁₁ cells depends on the level of GFP₁₋₁₀-NLS reaching into the nucleus, while level of GFP₁₋₁₀-NLS relies on the level of internalized cargo and the level of the internalized after successful endosomal escape. The western analysis in Fig.1e has shown us the amount of total internalization of GFP₁₋₁₀-NLS in the cell level is all about the same among different delivery system, hence the increased amount of GFP₁₋₁₀-NLS in the nucleus could be attributed to enhanced delivery (endosomal escape) at the level of endosomal escape. Moreover, **the similar MFI of the HEK-293T treated with various GFP-related proteins validated that the similar efficiency of internalization was presence among different delivery system, although addition of INF7 resulted in a slight increase of MFI (Please see above or revised Fig.1h).** To avoid this confusion, we have integrated above results and corresponding description into the revised MS (Page 6, Line 134-140).

3) One serious problem with almost all protein delivery assays is a lack of absolute measures of delivery. For example, Figures 1c and 1d show mean fluorescence intensity as a ratio of intensity observed with various constructs, to that observed with a construct that delivers little to no GFP₁₋₁₀. **Relative MFI is the ratio of a measurement over a small number.** The reader of this paper needs to know what MFI=10 means in terms of absolute amounts. **One could lyse cells, and use absolute fluorescence of the lysate compared to standard solutions of known concentrations of GFP to determine the absolute amount of GFP1-10 delivered.** But given the stochastic nature of delivery (Supplemental Figure 1) a microscopic technique might be better suited. This problem of quantitation is even worse for enzymes because their effect is greatly amplified.

Re: Thanks for your valuable comments. We agree with you about the need for measurement of absolute amounts of cargo delivered into cells, in particular the Fig 1 and 2, in which the relative MFI was used as index of delivery efficiency. In other words, we should show readers how much GFP were internalized into every cell (i.e., inside and outside of endosomes) and how much GFP₁₋₁₀-NLS escaped from the endosomes and localized into nucleus (i.e., endosomal escape) after treatment of various constructs.

I. The establishment of standard curve for full length GFP and GFP₁₋₁₀-NLS

The standard curve for GFP was established by directly analyzing the fluorescence intensity of GFP protein of known concentration (Please see below or new supplementary Fig.10a). As for the establishment of the standard curve for GFP₁₋₁₀-NLS, the progress curves for complementation of GFP₁₋₁₀-NLS and GFP₁₁ *in vitro* was investigated firstly. We

found that the fluorescence intensity starts to increase at 20 min after complementation starts, and reached plateau period at 4 hours (Please see below or new supplementary Fig.10b). Accordingly, the standard curve of fluorescence intensity vs.GFP₁₋₁₀-NLS concentration was identified at 4 hours after complementation starts(Please see below or new supplementary Fig.10c).

II. Quantitative determination of GFP inside the cell

The HEK-293T were treated with 1 μ M of GFP-related proteins in 1 mL serum-free DMEM for 3 h at 37 °C, followed by the medium containing proteins was aspirated, and the cells were washed 3 times with DMEM containing 10 U/mL heparin. As you suggested, cells were counted (1×10^5 cells), harvested and lysed, then the amount of full-length GFP in lysate was quantified by measuring its fluorescence and converting to intracellular protein concentration using corresponding standard curve. Intracellular protein concentration was calculated as described in previous study (Erazo-Oliveras et al. (2014) Nat Methods, 11: 861-867.). We demonstrated that approximately 57.6 μ M of GFP was present inside HEK-293T cells treated with TINNeL-GFP. Notably, consistent with the increment of MFI (fold increase) in FACS analysis, the amount of internalized cargo by TINNeL system was about 4-fold higher than that of TINNe and TAT peptide (Please see below or new supplementary Fig.10d).

III. Quantitative determination of GFP₁₋₁₀-NLS in the nucleus

Likewise, HEK-293T-GFP₁₁ were treated with 1 μ M GFP₁₋₁₀-NLS-related proteins in 1 mL serum-free DMEM for 3 h at 37 °C. At 12 h post incubation onset, the medium containing proteins was aspirated, and the cells were washed 3 times with DMEM containing 10 U/mL heparin. Cells were counted (1×10^5 cells), harvested and lysed, then the amount of GFP₁₋₁₀-NLS in lysate was quantified as described in GFP section. Of note, considering the leakage of entrapped GFP₁₋₁₀-NLS from endosome during cell lysis would intervene the quantitative determination of GFP₁₋₁₀-NLS in the nucleus, all steps including cell lysis and the measurement of fluorescence emission intensity was accomplished within 10 min (*in vitro* fluorescence intensity starts to increase from 20 min post mixing)(Please see below or new supplementary Fig.10b).On the basis of this assay, it was demonstrated that the amount of cytosolic delivery by TINNeL system is about 13-fold higher than that of TAT peptide only (Please see below or new supplementary Fig.10e), and similar increase was observed in the FACS analysis.

IV. Identification of fraction of cytosolic delivery

Since the similar amount of internalization between GFP and GFP₁₋₁₀-NLS mediated by same delivery peptides (Please see below or new supplementary Fig.10f), we provided objective quantitative insights into the process of TINNeL mediated-endosomal escape (the fraction of internalized cargo that reached the nucleus), which suggested that TINNeL induced at least 30% amount of total cellular internalization escaping from endosomes, remaining the same as TINNe, while only 10 % in that of T-GFP₁₋₁₀-NLS (Please see below or new supplementary Fig.10g). This indicates that in comparison with TINNe, TINNeL mediated-higher level of absolute amount of escaping-cargo mainly relies on enhancement of total internalization induced by its Leu zipper module.

New supplementary figure. 10 The determination of absolute amount of protein delivered into cells or in the nucleus mediated by various delivery system. **a** The standard curve of fluorescence intensity vs. GFP concentration was established. The fluorescence emission intensity of 100 μL aliquots containing 1.56 to 50 pmol of GFP were measured. **b** Progress curves for complementation of GFP₁₋₁₀-NLS of known concentration. 50 μL aliquots containing 0.78, 3.13 and 12.5 pmol of GFP₁₋₁₀-NLS were mixed with 50 μL aliquots containing 100 pmol GFP₁₁ peptides to start complementation, and fluorescence emission intensity was measured at indicated time point. **c** The standard curve of fluorescence intensity vs. GFP₁₋₁₀-NLS concentration was established. 50 μL aliquots containing 0.39 to 12.5 pmol of GFP₁₋₁₀-NLS were mixed with 50 μL aliquots containing 100 pmol GFP₁₁ peptides to start complementation, and fluorescence emission intensity was measured at 4 h post mixing. **d** Intracellular GFP concentration (left panel) and relative MFI of HEK-293T cells treated by 1 μM T-, TINNe-, or TINNeL-GFP in serum-free DMEM (1 mL). **e** Intracellular GFP₁₋₁₀-NLS concentration (left panel) and relative MFI of HEK-293T GFP₁₁ cells treated 1 μM T-, TINNe-, and TINNeL-GFP₁₋₁₀-NLS in serum-free DMEM (1 mL). **f** Immunoblot analysis of level of GFP and GFP₁₋₁₀-NLS (1 μM) delivered by TAT. **g** The identification of the fraction of internalized cargo that reached the nucleus based on various delivery systems. Results in a-e shown are means ± s.e.m, n = 3 for each group. **** P < 0.0001, and NS, no significant difference.

With regard to relative MFI (fold increase), please allow us to emphasize that it is obtained by MFI of total cells (not a small number) treated with indicated proteins divided by that of total cells treated with corresponding cargo protein only. This method is widely recognized and used in the CPP-based researches (Milech et al. (2015) Sci Rep, 5: 18329;Lonn, Kacsinta et al. (2016) Sci Rep, 6: 32301;Kauffman, Guha et al. (2018) Nat Commun, 9: 2568;Evans et al. (2019) Nat Commun, 10: 5012.). **Therefore, we reasoned that "Relative MFI" may be preferable index for readers who want to know how about our system/strategy in comparison with previous study.** Nevertheless, as you suggested, in our revision, based on the absolute amounts of delivered-cargo and its corresponding relative MFI, it was identified the one unit of relative MFI of GFP-related proteins treated cells means 1.4 μM GFP inside the cells (Please see below or new supplementary Fig.11a), while in Split-GFP assay, one unit means about 0.4 μM GFP₁₋₁₀-NLS escaped from endosome and located into nucleus (Please see below or new supplementary Fig.11b).

New supplementary figure. 11 The identification of intracellular GFP concentration(a) or GFP₁₋₁₀-NLS concentration(b) relative to one unit of relative MFI. The value of each dot was obtained by that protein concentration divided by its corresponding relative MFI analyzed by FACS. Results shown are means \pm s.e.m, n = 3 for each group. NS, no significant difference.

These results have been described briefly in the main text of revised MS (Page 9, Line 208-228) (M&M, Page 22, 535-543), while detailed information was included in revised supplementary information file (Page 11-15).

4) The writing syntax and sentence structure is very poor throughout the manuscript. The revised paper will need significant editing and polishing of the English. To provide one example, the opening paragraph of the discussion has several sentences that are nonsensical at present.

Re: Thanks for your helpful suggestion. The revised MS has now been proofread by a native English speaker doing active research in protein delivery, and the language has been edited and polished by **Springer Nature Author Services** before resubmission.

5) **The paper contains a number of statements that are essentially unsupported speculation.** I feel that they should be removed if they cannot be supported by some evidence from the literature or from these experiments. Such speculations include:

Line 119-120 “The CPP dimer might engage with the cellular membrane through multivalent interactions, but with, serum albumin in a monovalent manner; thus, the competitiveness of the serum is alleviated.”

Lines 253-256 ” Unexpectedly, endosomal escape efficiency of this dimer is quite low than observed at co-incubation approach of dfTAT30. The reason for this is unclear, it is possible that the effect of cargo molecules on the interaction between TAT with BMP (bis(monoacylglycerol)phosphate).”

Re: Thanks for your valuable suggestions. With respect to the first speculation, as suggested by Reviewer 2#, the interaction between T-GFP or TL-GFP with BSA was verified using HPSEC, and binding between BSA with both T-GFP and TL-GFP were observed. In addition, the delivery of T-GFP was dramatically decreased (61.3%) in the presence of 50 g/L BSA, while the delivery of TL-GFP was not significantly changed (Fig.2b). Therefore, interaction between CPP and negatively charged molecules such as BSA could decrease the delivery efficiency of monomeric CPP, but not the dimer. **Even though, we cannot identify whether both proteins associate only one molecule of albumin due to the limited resolution of HPSEC analysis. Therefore, revised this statement (Page 6, Line 142-156; Page 8, Line 172-176).**

Regarding the second statement, we agreed with reviewer's comment, and removed this speculation in the revised MS as suggested.

6) The animal experiments are very impressive and add a lot to this work. Again, we lack **any quantitative measure of delivery.** Is there a way to determine how much enzyme must be delivered to prevent the acute damage that drives the observed phenomenon?

Re: Thanks for your helpful comments. We agree that quantitative measurement of delivered Ppm1b in various tissues (e.g., liver) would be helpful for us to better characterize the efficacy of our platform. Due to difficulty of determination (e.g., lack of reliable and highly sensitive ELISA assay) and complexity of metabolism *in vivo*, currently it was technically challenging for us to quantify the absolute amount of Ppm1b *in vivo*. **Nevertheless, to solve your concerns as much as possible, we rather roughly compared relative delivery efficiency by two indirect approaches: tracking fluorescence dye-labelled proteins (e.g, eTAT-Ppm1b-Cy5), and immunohistochemistry.**

For tracking fluorescence dye-labelled proteins, twenty-four hours after intravenous administration with equimolar dose of eTAT-Ppm1b-Cy5, TAT-Ppm1b-Cy5 or Ppm1b-Cy5, all mice were euthanized(n=3), their main organs were collected, and the fluorescence intensity was analyzed. eTAT-Ppm1b showed higher fluorescence intensity than T-Ppm1b in multiple organs with different extent, confirming its excellent delivery performance *in vivo* (Please see below or revised Fig.3f and 3g). Moreover, consistent with the results of *ex vivo* imaging, the relatively quantitative analysis by using immunohistochemistry (Please see below or revised Fig.3h), as you suggested (**Comment 7**), validates the high efficacy of eTAT-based delivery. These results and relevant description have been included in the revised MS (Page 11, Line 262-277).

Fig. 3 eTAT-Ppm1b suppresses TNF-induced necroptosis in vitro and in vivo. **f, g** Distribution of intravenously administered Cy5 labelled-Ppm1b-related proteins in the female BALB/c mice. Representative microscope images (**f**) and fluorescence intensity (**g**) in the indicated organs of the BALB/c mice intravenously administered with Ppm1b-Cy5, TAT-Ppm1b-Cy5 and eTAT-Ppm1b-Cy5 (n=3). Twenty-four hours post-administration, the mice were sacrificed, and fluorescence imaging of each organ was performed. **h** Five hours after injection with indicated constructs or PBS, the tissues were harvested and prepared as paraffin slides. The nuclei were stained with DAPI, and the delivered Ppm1b were detected using anti-6×His IgG as the primary antibody and Alexa 647-conjugated goat anti-mouse IgG as the secondary antibody. The fluorescence was captured via fluorescence microscopy. Yellow boxes in the images indicate magnified region. The data in **g** are expressed as means ± s.e.m. * P<0.05, ** P<0.01, *** P<0.001, **** P < 0.0001 and NS, no significant difference.

Likewise, due to lack of appropriate approach of absolute quantification, it is also a big technical challenge for us the identification of how much enzyme must be delivered to cells in vivo to prevent the acute damage. Even though, we sought to answer this question by identify how much eTAT-Ppm1b were required to achieve therapeutic effect. In this revision, we have investigated the impact of different dose of eTAT-Ppm1b on APAP induced-ALF in the mouse model. The administration of 5 nmol eTAT-Ppm1b protein twice in 2 h and 6 h post-APAP injection almost fully inhibited the drug-induced the elevation of the level of ALT and AST in the serum, thus the mouse was intravenously administrated with 2 to 6 nmol eTAT-Ppm1b twice in 2 h and 6 h post-APAP administration to identify the minimum dose required to achieve effect. When at the concentration below 5 nmol, the level of ALT and AST elevated obviously. We thus concluded that the effective inhibition on APAP induced-ALF was observed when twice dose of 5 nmol (~15 mg/kg) eTAT-Ppm1b adopted

at least. This effective dose was safe for mouse as no significant change of ALT and AST in serum in our research was observed (Please see below or new supplementary Fig. 11). Moreover, this dose was moderate level for CPP-based drug via i.v. injection, as indicated in the previous study (i.e., the median lethal dose of the CPP-oligomer conjugates administered iv was between 210 and 250 mg/kg) (Cai et al. (2006) Eur J Pharm Sci, 27: 311-319; Jarver et al. (2010) Trends Pharmacol Sci, 31: 528-535.) (Please see below or new supplementary Fig. 16). These results and relevant description have been included in the revised MS (Page 13, Line 300-305)

New supplementary figure. 16 Identification the minimum dose of eTAT-Ppm1b required to achieve therapeutic effect in female BALB/C. **a** Wild-type female BALB/c mice were intravenously administrated with 2 to 6 nmol eTAT-Ppm1b twice in 2 h and 6 h post-APAP administration. **b-c** Serum was collected at 12 h post APAP administration, then activities of **(b)** ALT and **(c)** AST were determined by routine clinical assays using commercial kits. Results shown are means \pm s.e.m, $n = 7$ mice for each group. **** $P < 0.0001$, and NS, no significant difference.

7) Can the authors use the Cy5 fluorescence to determine the amount of enzyme delivered to cells? Can fluorescence microscopy be done on tissue sections to directly demonstrate and quantify cargo delivery to cells in vivo.

Re: We greatly appreciate your valuable suggestion. With regard to the determination of amount of enzyme delivered to L929 using Cy5 fluorescence, **we reasoned that, as described in the section of absolute measurement of GFP, direct measurement of fluorescent-tagged proteins can't specifically discriminate between endosomal escape and endosomal entrapment. In other words, it cannot reflect the amount of cytosolic delivery.** Nevertheless, inspired by this comment and Reviewer 3# 's comment (the last one), in this revision, to investigate the efficacy of eTAT system when in the delivery of Ppm1b, we compared the relative level of Ppm1b delivered by eTAT and other control system at two key time point by western blotting. The results of 30 min post treatment could indicate the level of total cellular internalization (cytosolic delivery+

endosomal entrapment), while that of 12 h post treatment would indirectly reflect the level of cytosolic delivery (not all the cytosolic delivery), due to the entrapped proteins in the endosomes would be degraded quickly in the lysosome. As shown in Revised Fig.3a or see below, at 30 min post treatment, the level of Ppm1b delivered by eTAT in L929 was highest and that of all other TAT-based constructs are about same except Ppm1b alone. When 12 h post incubation onset, the western blotting demonstrated that higher level of Ppm1b (cleaved form) was detected in the cells treated with eTAT-Ppm1b than TINNe-, and no Ppm1b protein could be detected in cells treated with other TAT-based constructs. The efficient cytosolic delivery of eTAT-Ppm1b at 12 h post incubation onset was attributed to enhanced cellular uptake and more efficient endosomal escape. **In general, these data suggest the eTAT show the highest efficiency for Ppm1b delivery.** Now, we have integrated above results and corresponding description into the revised MS. (Page 10, Line 237-241).

Revised Fig. 3 eTAT-Ppm1b suppresses TNF-induced necroptosis *in vitro* and *in vivo*. a Immunoblot analysis of level of Ppm1b in L929 cells treated with Ppm1b-related proteins (1 μ M) at indicated time point.

With regard to *in vivo* delivery, we performed immunostaining on paraffin sections of caecum and liver (two organs in which Ppm1b-related proteins were mainly distributed), at 5 h following a single injection of eTAT-Ppm1b and other control constructs. Our results showed that significant red fluorescence signal was observed in the caecum and liver of eTAT-Ppm1b-treated mice and the intensity was stronger than that of T-Ppm1b. On higher magnification, Alexa-647 signal of eTAT-Ppm1b was bright and diffused in the cytosol of tissue cells, confirming *in vivo* intracellular protein delivery (Please see below or revised Fig. 3h). **Collectively, our results demonstrate that eTAT can deliver functional proteins Ppm1b more efficiently to cells *in vivo*.** We have integrated above results into the revised manuscript, and greatly appreciate the reviewer's constructive suggestion (Page 12, Line 269-276).

Revised Fig. 3 eTAT-Ppm1b suppresses TNF-induced necroptosis *in vitro* and *in vivo*. **h** Five hours after injection with indicated constructs or PBS, the main tissues were harvested and prepared as paraffin slides. The nuclei were stained with DAPI, and the delivered Ppm1b were detected using anti-6×His IgG as the primary antibody, and Alexa 647-conjugated goat anti-mouse IgG as the secondary antibody. The fluorescence was captured via fluorescence microscopy. Yellow boxes in the images indicate magnified region.

Reviewer #2 (Remarks to the Author):

Yu et al. report on a novel CPP(TAT)-based intracellular protein delivery system (eTAT) consisting of a CPP, PMAP, endosome-specific protease sites and a leucine zipper, showing enhanced cytosolic protein, phosphatase 1B (Ppm1b), delivery efficiency and serum tolerance in vitro and in vivo without toxicity, suppressing the TNF- α -induced systemic inflammatory response and acetaminophen-induced acute liver failure in the mouse model.

They provide a strategy to facilitate the development of CPP-based delivery platform on proteins for intracellular targets. The work is well performed and timely.

However, my concern is that it is difficult to see clear advantage of this method as compared to previously published protein delivery systems, e.g. TAT peptide and multiple others. **I suggest to carry out additional studies in order to confirm such advantage.**

Re: We appreciated your positive summary and comments of our study. We are sorry for not clearly describing the advantages in our initial submission, this point touches one the same elements as raised by Reviewer #3 (point 1). Now we conclude the advantages of our delivery strategy and novelty of our study (Please see Response to the first comment of Reviewer #3). Briefly, one major finding in our study is providing a multi-modular CPP-based strategy for protein delivery to overcome the persistent drawbacks in the CPPs-based application. Accordingly, we choose the most widely used CPP, the translocation peptide derived from the HIV transactivator of transcription (TAT), as the prototypical CPP (T-cargo); thereby TAT-cargo is one of critical control delivery systems.

As reviewers suggested, to robustly confirm and clearly provide the advantages of our method, including enhanced delivery efficacy and serum tolerance in vivo, as compared to TAT-based delivery system or it containing other modules (e.g., INF7), we did additional experiments as listed below. We hope these additional data supplemented in the revision can enable our MS to win your satisfaction.

I. Refer to Revised Fig. 2 (or see below)

The data suggest that in compared with TAT-cargo, the dimerization of CPP-fused cargo mediated by leucine-zipper exhibits enhanced serum tolerance, mainly achieved by weakening the competitiveness of serum protein albumin, **indicating that oligomerization of CPP-fused cargo was critical for its serum tolerance.**

Revised Fig. 2 Dimerization of CPP-fused proteins enhances endocytosis and serum tolerance. a The MFI of T-GFP treated HEK-293T cells (5 μ M) in the free-serum DMEM, DMEM containing BSA (50g/L) or 100% FBS. **f** The MFI of TL-GFP treated HEK-293T cells (5 μ M) in the free-serum DMEM, DMEM containing BSA (50g/L) or 100% FBS. Percentages presented in the columns were obtained by dividing the MFI of treated cells in the given condition by the MFI of treated cells in free-serum DMEM. The data are expressed as means \pm s.e.m. * $P < 0.05$, ** $P < 0.01$, *** $P < 0.001$, **** $P < 0.000$, and NS, no significant difference.

II Refer to revised Fig.3 (or see below)

The WB analysis demonstrated that the eTAT showed the highest level of Ppm1b delivery in L929 cell line, in compared with T-, TI- as well as TINNe-based system, **suggesting the stronger delivery performance of eTAT system in vitro and for which all of four functional modules were responsible.**

Revised Fig. 3 eTAT-Ppm1b suppresses TNF-induced necroptosis in vitro and in vivo. a Immunoblot analysis of level of Ppm1b in L929 cells treated with Ppm1b-related proteins (1 μ M) at indicated time point.

III. Refer to Revised Fig. 3 (or see below)

We demonstrated that eTAT-Ppm1b exhibited higher level of distribution in the multiple tissues, such as caecum, liver, of the BALB/C mouse based on the approach of the *ex vivo* imaging and immunohistochemistry, suggesting eTAT system is more potent than TAT in protein delivery *in vivo*.

Revised Fig. 3 eTAT-Ppm1b suppresses TNF-induced necroptosis in vitro and in vivo. **f, g** Distribution of intravenously administered Cy5 labelled-Ppm1b-related proteins in the female BALB/c mice. Representative microscope images (**f**) and fluorescence intensity (**g**) in the indicated organs of the BALB/c mice intravenously administered with Ppm1b-Cy5, TAT-Ppm1b-Cy5 and eTAT-Ppm1b-Cy5 (n=3). Twenty-four hours post-administration, the mice were sacrificed, and fluorescence imaging of each organ was performed. **h** Five hours after injection with indicated constructs or PBS, the tissues were harvested and prepared as paraffin slides. The nuclei were stained with DAPI, and the delivered Ppm1b were detected using anti-6×His IgG as the primary antibody and Alexa 647-conjugated goat anti-mouse IgG as the secondary antibody. The fluorescence was captured via fluorescence microscopy. Yellow boxes in the images indicate magnified region. The data in g are expressed as means ± s.e.m. * P<0.05, ** P<0.01, *** P<0.001, **** P < 0.0001, and NS, no significant difference.

Reviewer #3 (Remarks to the Author):

Authors have designed and refined Tat-peptide-based vector for delivering into cells protein cargoes covalently coupled to peptide. First, the widely used CPP was supplemented with PMAP sequence, and two protease cleaving sites for facilitating release of cargo proteins from endosomal entrapment. Later the dimerisation-inducing motif was included in peptide vector to increase the uptake by cells and reduce sensitivity towards serum albumin that is known to inhibit Tat-mediated delivery into cells.

The efficiency of elaborated eTAT vector was verified by delivering GFP and its split version into model cell-lines. Finally, eTAT was used for delivery of Pmp1b to suppress TNF-induced necroptosis *in vitro* and *in vivo*. Moreover, eTAT-Pmp1b also increased survival of mice in acetaminophen-induced acute liver failure model.

Re: We deeply appreciate your careful reading of our MS and the constructive comments provided. We have modified the MS accordingly.

Major comments:

1. Authors claim that the used strategies for improving Tat-peptide mediated delivery are novel. However, higher delivery efficacy of multivalent CPPs compared to monovalent peptide has been earlier demonstrated by **several laboratories e.g. of Futaki and Gariepy. Their works have to be referred to in introduction.** The idea of using cleavage by protease for activation of CPP construct has been repeatedly exploited in the field of CPPs, and **was pioneered by R. Tsien group, who perhaps also deserves mentioning.**

Re: Thanks for your valuable comment. As you correctly point out, the multivalent CPPs (MCPs) and the strategy of proteolytic cleavage activated-delivery were reported in previous publications. As suggested, the relevant references have been cited in the revised MS. **However, there is still significant novelty in the current study in comparison to related publications as described below. To solve your concerns, we have added clarifying statements in the main text of our revised MS to better describe our innovations with respect to what's been done previously.**

I. First of all, the most significant novelty of this paper relies on the novel multi-modular CPP based-strategy for protein delivery. The four functional modules, constituted to one chimeric peptide delivery system, were all critical for the excellent delivery efficacy *in vivo* and *in vitro*. The sentences relevant to this novelty have been added in the introduction section of revised MS (Page 4, Line 71-78).

II. Novel utilization of proteolytic cleavage-mediated activation. As you mentioned, the strategy of proteolytic cleavage activated-delivery has been widely utilized in the field of CPPs, after being pioneered by the group of Roger Y. Tsien for tumor imaging before (Jiang et al. (2004) Proc Natl Acad Sci U S A, 101: 17867-17872; Aguilera et al. (2009) Integr Biol (Camb), 1: 371-381.). For instance, proteolytic cleavage mediated by extracellular proteases (i.e., matrix metalloproteinases, MMP) specifically restores the penetrating activity of CPP moiety in the vicinity of tumor cells. However, these researches mainly focused on improvement of specificity in CPP-based delivery (He et al. (2016) J Control Release, 240: 67-76.). The strategy of proteolytic cleavage-based activation in our study, directed toward other intracellular enzymes, endosome-localized proteases (e.g., CTSL,

furin), and was utilized to facilitate the endosomal escape. Due to the different intentions, the study of R. Tsien group was not mentioned in the original submission, now the relevant references have been added to the discussion section of revised MS as suggested (Page 14, Line 332-339).

III. Novel approach for multivalent CPP-fused proteins formation. Many previous studies reported various strategies to prepare MCPP(Sung et al. (2006) *Biochim Biophys Acta*, 1758: 355-363.), such as branched peptides system(Futaki et al. (2002) *Biochemistry*, 41: 7925-7930.) and attaching a protein oligomerization domain (e.g., p53tet) (Kawamura et al. (2006) *Biochemistry*, 45: 1116-1127.) to CPPs. In the present study, we prepared multivalent CPP-fused protein by using leucine zippers motif for the first time. After purification of the TAT-GFP containing leucine zippers peptides (TL-GFP), it could self-assemble into a dimeric CPP–cargo. Similar dimer formation has been observed when cargo was GFP₁₋₁₀-NLS or Ppm1b as well, indicating that leucine zipper mediated-dimerization would be a convenient strategy to generate multivalent CPPs. The relevant researches (including that of Futaki and Gariépy) have been referred in the introduction section of revised MS as you suggested (Page 3, Line 66-70).

IV. Novel solution to serum intolerance: In the previous study, MCPP was usually employed to increase the cellular uptake of CPPs (Kalafatovic and Giralt (2017) *Molecules*, 22: .), limited researches focus on its impact on serum tolerance. **In the current study, we provided a proof-of-concept that the oligomerization of CPP-fused cargo was critical for its serum tolerance, which is probably achieved by weakening the competitiveness of serum protein albumin.** The sentences relevant to this novelty have been included in the discussion section of revised manuscript (Page 15, Line 359-364).

2. Fig. 1a, a scheme of recombinant cell-penetrating protein, does not reveal where the polyhistidine tag that is used for purification of protein, is placed in the sequence. Please add! All data in manuscript suggest that histidine tag is still present on all used protein constructs, and it is not removed. **Histidine residues are known to serve as a proton sponge and facilitate escape from endosomes, was its impact considered?**

Re: Thanks for your valuable comment. Firstly, we apologies for the omission of symbol of polyhistidine tag (His-tag) in original submission, which has been added for all constructs now. Due to the presence of His-tag in all recombinant proteins, thus we have not considered its effect before. As you correctly indicated, histidine residues are known to serve as a proton sponge and facilitate escape from endosomes, thus poly-histidine sequences have been used as motifs to improve endosomal escape in trans-delivery (co-incubation) of gene (Lo and Wang (2008) *Biomaterials*, 29: 2408-2414.) or ribonucleoproteins (Del'Guidice et al. (2018) *PLoS One*, 13: e0195558.). However, up to we know, there was ever no study focusing on its impact on endosomal escape when fused to CPP-cargo proteins. Based on the previous literatures by Gariépy group(Mohammed et al. (2012) *J Control Release*, 164: 58-64.) and Doi group(Sudo et al. (2017) *J Control Release*, 255: 1-11.) aimed at the function of new PMAPs, the His-tag was present on all used constructs in their study as well, which may suggested its role in endosomal escape could be ignored. Nevertheless, to address your concern, we have now included new data to assess the effect of His-tag on the endosomal escape using our novel Split-GFP assays (Please see Reviewer 1# point 1). The TINNe-GFP₁₋₁₀-NLS without six histidine residues

(6H) was constructed and purified (Please see Fig. R2a and R2b below). Then, the HEK-293T-GFP₁₁ cells were treated by 5 μ M TINNe-GFP₁₋₁₀-NLS without HIS-tag (TINNe-GFP₁₋₁₀-NLS-6H (-)) and TINNe-GFP₁₋₁₀-NLS (TINNe-GFP₁₋₁₀-NLS-6H (+)). Our data show that there is no significant difference between MFI of HEK-293T-GFP₁₁ treated by two proteins (Please see Fig. R2c below). Collectively, this experiments according your suggestion show that the presence of His-tag would not intervene the assessment of endosomal escape efficiency in current study.

Fig.R2 The impact of fused-HIS-tag on efficiency of endosomal escape. a Schematic diagram of TINNe-GFP₁₋₁₀-NLS-6H (-) and TINNe-GFP₁₋₁₀-NLS-6H (+). **b** SDS-PAGE analysis of TINNe-GFP₁₋₁₀-NLS-6H (-) at different purification steps. Sample in lane six was used for further experiment. **c** The MFI of the HEK-293T cells treated with by 5 μ M TINNe-GFP₁₋₁₀-NLS-6H (-) or TINNe-GFP₁₋₁₀-NLS-6H (+). Results in c shown are means \pm s.e.m: NS, no significant difference.

3. Fig 1h. The intensity of GFP signal is substantially higher for TI-GFP compared to T-GFP after incubation MA104 cells with the respective protein constructs. The efficiency of cationic CPPs is dependent on the charge of peptide (R and K in sequence). Addition of INF7 to TAT sequence reduces the charge of targeting sequence from 8 to 4, which typically strongly impairs internalisation of peptide/protein construct. **Here the fluorescence microscopy shows opposite – protein carrying the transducing sequence with lower positive charge is taken up by cells more efficiently than one with higher charge. Please explain this discrepancy in discussion part.**

Re: We appreciate this valuable comment. In Fig.1h of original MS, the fusion of INF7 to TAT sequence, which reduces the net charge of delivery sequence, conversely resulted in the enhancement of intracellular uptake. Actually, similar phenomenon was observed in previous researches (Neundorf et al. (2009) Pharmaceuticals (Basel), 2: 49-65; Lee et al. (2011) Biochim Biophys Acta, 1810: 752-758; Sudo, Niikura et al. (2017) J Control Release, 255: 1-11; Patel et al. (2019) Sci Rep, 9: 6298.). **Among this, the structural analysis in the research by Wöfl group (Neundorf, Rennert et al. (2009) Pharmaceuticals (Basel), 2: 49-65.) provided a probable mechanism behind this observation: the attachment of the fusion peptide (i.e., PMAP) had profound implications on the whole conformation of the peptide (CPP-PMAP), which might promote the interaction between CPPs and membrane components.** Moreover, some relevant literatures may

help us explain why constructs carrying the transducing sequence with lower positive charge is taken up by cells more efficiently than that of higher charge: firstly, positively charged residues R within CPPs indeed play a critical role in the uptake process, yet it has been demonstrated that guanidino moieties of R in delivering peptide played a greater role in facilitating cellular uptake than either the charge or the backbone structure (Mitchell et al. (2000) J Pept Res, 56: 318-325.); secondly, other factors, such as physical nature of fused-cargo also contribute to the extent of internalization. For example, two transducing sequence with different positive charge, when fused to cargo of small molecule TAMRA or GFP₁₁ peptide, they exhibit reverse rank order of delivery efficiency (Kauffman, Guha et al. (2018) Nat Commun, 9: 2568.).

Additionally, inspired by this comment, in order to better characterize the internalization efficiency various GFP-related recombinant proteins in the HEK-293T, the additional FACS analysis was included in the revised MS. Consistent with the observation of imaging, FACS analysis showed that the addition of INF7 enhanced, but slightly, the internalization efficiency of T-GFP (Please see below or revised Fig.1h).

Please show more than 1 or 2 cells in new version of Fig 1h.

Re: We revised the Fig 1h, in which there were at least representative five cells. (Please see below or revised Fig.1g).

In addition, we have integrated above results into the revised MS (Page 6, Line 134-140), and explain reasons behind this observation in discussion section (Page 14, Line 339-348).

Revised Fig. 2 Dimerization of CPP-fused proteins enhances endocytosis and serum tolerance. g

The representative microscope images of cytosolic fluorescence distribution of the MA104 cells treated with different GFP-related proteins as indicated (Upper panel: scheme diagram of GFP-related constructs). Proteins were used at a concentration of 1 μ M. **h** MFI (left panel) within total HEK-293T cells and corresponding percentage of green fluorescence positive-cells (right panel) after the treatment of different GFP-related proteins (n=3). Proteins were used at a concentration of 1 μ M. Results in **h** shown are means \pm s.e.m: * $P < 0.05$ and NS, no significant difference.

4. Line 123 ... As analyzed by SDS-PAGE, T-GFP and TL-GFP are monomers and dimers respectively (Fig. 2b). **It is highly improbable that in SDS-PAGE the latter protein construct is in the form of dimer only. The Supplementary Figures 5, 6 and 8 show quite opposite.**

Re: Thanks for your comment. As we know, in the SDS-PAGE analysis, SDS would influence on the dimerization of leucine zipper-fused protein in different extent, depending on the physical nature of fused-cargo proteins and other motifs. We speculated that, when leucine zipper was fused to T-GFP, the formation of dimer is still enough stable in SDS-PAGE (without boiling), thus TL-GFP could be in the form of dimer only (or majorly). However, when fused to other proteins (e.g., T-GFP₁₋₁₀), leucine zipper induced-dimer formation may be easier to be affected by SDS, thereby, as shown in SDS-PAGE of original Supplementary figures. 5, 6 and 8, only a fraction of leucine zipper-fused protein keeping leucine zipper motif induced-homogeneous dimerization. This is why dimer formation was investigated by using SDS-PAGE and HPSEC in our study. Prompted by your comments, we added the HPSEC analysis of TL-GFP and TL-GFP₁₋₁₀ in this revision, which was missing in original submission (Please see below or revised Fig.2c, revised supplementary Fig. 7b).

Revised Fig. 2 Dimerization of CPP-fused proteins enhances endocytosis and serum tolerance. c Reduced SDS-PAGE (protein samples not boiled) and HPSEC analysis of purified T-GFP and TL-GFP; the Table shows the retention time and molecular weight of the samples.

Supplementary figure. 7 The TAT-GFP₁₋₁₀-NLS and its counterpart containing leucine zipper were analyzed by reduced SDS-PAGE and HPSEC. b The HPSEC retention time of T- and TL-GFP in the PBS.

5. Line 191 ... Twenty-four hours after administration, eTAT-Ppm1b was mostly distributed in the caecum ...

This is rather surprising observation. Has that been shown for other CPPs or CPP-protein constructs earlier? Please comment on this in discussion.

Re: We appreciate your excellent comment. It is interesting observation for us, in this revision, we sought to further investigate whether or not eTAT peptide induce caecal distribution of eTAT-Ppm1b.

Initially, we measured florescence intensity in multiple organs of mouse injected with Cy5-labelled constructs, it was suggested the Ppm1b-Cy5, T-Ppm1-Cy5 and eTAT-Ppm1-Cy5 all mostly distributed in the caecum (Please see Fig.R3a and 3b below), which was never observed in previous CPP-based researches up to we know. To clarify whether eTAT or Ppm1b proteins would target to caecum, and resulting in caecum distribution, we delivered Cy5-labelled eTAT-GFP to BALB/C mice by intravenous instillation. We demonstrated that there is extremely lower Cy5 signal observed in the caecum of eTAT-GFP treated mice, while the similar level of florescence intensity was detected in the corresponding organs of eTAT-Ppm1b-treated mouse. (Please see Fig.R3a and 3b below).

Moreover, we further identify these observations using immunohistochemistry. The female BALB/C mice was intravenously administrated with Ppm1b, T-Ppm1b, eTAT-Ppm1b and eTAT-GFP. At 5 hours after treatment, the tissues were harvested and prepared as paraffin section. The delivered Ppm1b were detected using anti-6×his IgG as the primary antibody and Alexa 647-conjugated goat anti-mouse IgG as secondary antibody. Consistent with what was observed in ex vivo imaging. Our results showed that significant red fluorescence signal was observed in the various tissues, especially in the caecum, of eTAT-Ppm1b-treated mice and the intensity was stronger than that of T-Ppm1b (Please see below Fig.R4c). Moreover, there is extremely lower Alexa 647 signal was detected in the caecum of eTAT-GFP treated mice (Please see below Fig.R4c). **These results suggest that the Ppm1b, not eTAT or any modules within it, induce the extraordinary caecum distribution of eTAT-cargo. The reason behind this phenomenon is clearly an area for our future research.**

We have integrated above results into the revised MS followed the suggestion of the reviewer (Page 12, Line 277-282), added this information in the discussion section and clearly stated that further experiments are needed to verify the reasons (Page 16, Line 387-391).

Fig.R3 In vivo protein delivery efficiency of eTAT system. **a, b** Distribution of intravenously administered Cy5 labelled-Ppm1b-related proteins in the BALB/c mice. Representative microscope images **(a)** and fluorescence intensity **(b)** in the indicated organs of the BALB/c mice intravenously administered with Ppm1b-Cy5, TAT-Ppm1b-Cy5, eTAT-Ppm1b-Cy5 and eTAT-GFP-Cy5 (n=3). Twenty-four hours post-administration, the mice were sacrificed, and fluorescence imaging of each organ was performed. **c** Five hours after injection with indicated constructs or PBS, the tissues were harvested and prepared as paraffin slides. The nuclei were stained with DAPI, and the delivered cargo proteins were detected using anti-6×His IgG as the primary antibody and Alexa 647-conjugated goat anti-mouse IgG as the secondary antibody. The fluorescence was captured via fluorescence microscopy. Yellow boxes in the images indicate magnified region. Results in **c** shown are means ± s.e.m: *** $P < 0.001$, **** $P < 0.0001$, and NS, no significant difference.

6. Line 234 ... the endocytosis of the cargo was mediated by the interaction between TAT and the cell membrane (Fig. 5a)

Does TAT peptide/motif interact directly with membrane? It has been repeatedly shown that proteins are involved in interaction and penetration of this CPP across membranes

Re: We apologize for these confusing sentences. The endocytosis of CPPs with cells is initiated by interaction with membrane components on the cell surface, including glycosaminoglycans (GAGs), protein receptors or phospholipids, not directly with membrane. To avoid this confusion, we have corrected these sentences into " **the endocytosis of the cargo was mediated by the interaction between TAT and membrane components on the cell surface (e.g., glycosaminoglycans)** " in the revised MS (Page 14, Line 322-323).

7. Discussion contains multiple factual errors and difficult-to-understand sequences. It requires revision with **focusing on obtained results** and leaving out ample putative developments of presented protein delivery vector.

Re: We appreciated you for this suggestion. The discussion has been rephrased to better summarize our major findings, as well explain the novelty and limitations of our method.

8. In multiple places the claims of authors are presented **without the respective reference/citation** (e.g. line 44-45, 50 etc.).

Re: We have checked all the references throughout the manuscript and greatly appreciate your valuable comment.

9. The manuscript contains many typos (e.g. line 33), citation errors (e.g. line 59, line 118 [18]) and sentences that are difficult to read (e.g. lines 24-28, 37-40, 250-254) or are left unfinished (e.g. lines 255-256).

Re: We greatly thank for your efforts paid on reviewing our MS. We have carefully checked the typos and grammar errors throughout the manuscript. In addition, the revised MS has now been proofread by a native English speaker doing active research in protein delivery, and the language has been edited by **Springer Nature Author Services** before resubmission.

Specific comments:

1. Line 52-53 ... After endocytosis mediated by CPP, the majority (approximately 90%) of the cargos are entrapped inside endosomes.

Does it mean that 10% liberates and is active? Please provide reference for 90%.

In general, one or two logs lower escape/activity is considered realistic.

Re: Thanks for your valuable comment. As you correctly pointed out, one or two logs lower escape/activity is realistic. In the revised MS, we amended the description to **"the majority (e.g., approximately 99 % when in the delivery of Cre recombinase) of the cargos are entrapped inside endosomes"** based on the review paper written by Dowdy group (Lonn and Dowdy (2015) Expert Opin Drug Deliv, 12: 1627-1636.), and added this reference in the revised MS as suggested (Page 3, Line 55).

2. Line 70-72 ... Therefore, we believe the eTAT system or its improved version composed of multifunctional chimeric peptide has potential to become a major enabling strategy for protein delivery in biological research and therapeutic application.

Exaggeration, considering how large quantities of protein construct (per kg) were required to achieve effect in mouse model.

Re: Thanks for your valuable suggestion, this point touches one the similar elements as raised by Reviewer 1 #. We sought to answer this question by using the APAP induced-ALF mouse model. Actually, in our preliminary study on TNF- α induced SIRS model (prevention model), we have demonstrated that the minimal dose of eTAT-Ppm1b achieving the 100 % survival ratio was 5 nmol (Data was not shown in original MS, please see Fig.R4 below) followed by the comparison of different Ppm1b-related constructs. Accordingly, the dose of 5 nmol (twice) was employed in the APAP induced-ALF model (therapy model).

In the revised MS, to identify minimal does of eTAT-Ppm1b required to achieve therapeutic effect, the mouse was intravenously administrated with 2 to 6 nmol eTAT-Ppm1b twice in 2 h and 6 h post-APAP administration. When at the concertation below 5

nmol, the level of ALT and AST elevated obviously. We thus concluded that the effective inhibition on APAP induced-ALF was observed when twice dose of 5 nmol (~15 mg/kg) eTAT-Ppm1b adopted at least. This effective dose was safe for mouse as no significant change of ALT and AST in serum in our research was observed (Please see below or new supplementary Fig. 17). Moreover, this dose was moderate level for CPP-based drug via i.v. injection, as indicated in the previous study (i.e., the median lethal dose of the CPP-oligomer conjugates administered iv was between 210 and 250 mg/kg) (Cai, Xu et al. (2006) Eur J Pharm Sci, 27: 311-319; Jarver, Mager et al. (2010) Trends Pharmacol Sci, 31: 528-535.) (Please see below or new supplementary Fig. 16). These results and relevant description have been included in the revised MS (Page 13, Line 299-304)

Figure R4. Identification the minimum dose of eTAT-Ppm1b required to achieve 100 % survival ratio in female BALB/C. Wild-type female BALB/c mice were intravenously administrated with 3 to 6 nmole eTAT-Ppm1b followed by TNF challenge. Mouse survival was monitored every hour for 40 h, with the results presented in a Kaplan–Meier plot, and a log-rank test was performed; n=8 mice for each group. * $P < 0.05$, **** $P < 0.0001$ and NS, no significant difference.

New supplementary figure. 16 Identification the minimum dose of eTAT-Ppm1b required to achieve therapeutic effect in female BALB/C. **a** Wild-type female BALB/c mice were intravenously administrated with 2 to 6 nmol eTAT-Ppm1b twice in 2 h and 6 h post-APAP administration. **b-c** Serum was collected at 12 h post APAP administration, then activities of **(b)** ALT and **(c)** AST were determined by routine clinical assays using commercial kits. Results shown are means \pm s.e.m: n = 7 mice for each group. **** $P < 0.0001$ and NS, no significant difference.

3. In Figure 1e type of actin?

The level of actin varies substantially between lines. **Please specify (in Materials and Methods) how the amount of protein that was applied to gel was equalised.**

Re: We apologies for this omission. β -actin was used as loading control, and we corrected the label "Anti-actin " to "Anti- β actin" in Fig.1e in the revised MS.

To equalize the amount of protein which was applied to gel, firstly, same number of (5×10^5 cells per well in 12-well plates) were seeded in cell culture plates. Next, after treatment with the conditions specified in the figure legends, all cells in the plate were detached, resuspended in PBS, and the cell numbers were determined and adjusted to 1×10^5 cells per 100 μ L lysis buffer for WB analysis, and equal volume was loaded to each lane. **Now we have specified this process in Materials and Methods as reviewer suggested (Page 21, Line 490-491, 492-495).**

In the Fig.1e, cells were treated with different GFP₁₋₁₀-related proteins at the same time, then collected at the indicated time point. Thus, the variation of the β -actin level between lines, especially 3 h/6 h with other time point, may be attributed to that anti- β -actin antibody was recovered from experiments in 0.5 h and 1 h. Nonetheless, due to similar level of β -actin showed in different lanes, so we reason that our conclusion could be supported by current data.

4. Figure 1f. Fluorescence of GFP in split assay peaks at 9 h time-point and **starts to decrease from there,** which occurs surprisingly quickly. Please comment this in discussion.

Re: Thanks for your valuable comment. As shown in Fig.1e, the western blotting analysis suggested that a fraction of complete GFP constituted of GFP₁₋₁₀ and GFP₁₁ have degraded between 9 h and 12 h post-incubation onset. Accordingly, fluorescence intensity of GFP decrease quickly from 9 h post-incubation onset. To further validate this observation, we have repeated this experiment using novel Split-GFP assay, and have updated the results in Fig. 1e. The new results suggest a similar pattern of kinetics of green fluorescence (Please see below or revised Fig.1f).

Fig. 1 Proteolytic removal of CPP-PMAP in the endosomes enhances the escape efficiency of protein cargos. **f** the MFI in the treated cells at different time points during the endonuclear split GFP assay. GFP₁₋₁₀-NLS-related proteins were used at a concentration of 1 μ M. The results shown are the means \pm s.e.m.; * $P < 0.05$, *** $P < 0.001$, **** $P < 0.0001$.

5. Lines 118-119 ... homodimerization capacity was employed to mediate the dimerization of the CPP-fused proteins.

This approach has been used with CPPs earlier. **Please add reference.**

Re: Thanks for your helpful suggestion. The relevant references have been cited in the revised MS.

6. Lines 119-121 ... The CPP dimer might engage with the cellular membrane through multivalent interactions, but with, serum albumin in a monovalent manner; thus, the competitiveness of the serum is alleviated.

What gives ground for this assumption? Has it been shown experimentally?

For verifying this assumption, **the authors are advised to analyse the elution of dimeric protein construct (after incubation with serum or not) from SEC column.** Comparison of that with elution profiles of monomeric from should reveal, whether both proteins associate only one molecule of albumin.

Re: This is an excellent advice. The suggested experiment has now been performed in the revised MS. It has been proposed previously that CPP could binds to negatively charged proteins in the serum, such as albumin (Kosuge et al. (2008) Bioconjug Chem, 19: 656-664.) (accounted for 35-50g/L in the serum), and which would suppress the CPPs-mediated endocytosis (Mueller et al. (2008) Bioconjug Chem, 19: 2363-2374.). But it was just speculation and there was still no direct evidence that CPPs could bind to BSA and thus suppress the CPPs mediated endocytosis. As you suggested, we have now verified the binding between TAT motif and BSA (bovine serum albumin), which is the richest components in serum, by using HPSEC (Please see below or new supplementary Fig. 6a). Initially, 25 μ M GFP or T-GFP were incubated with 75 μ M BSA individually (molar ratio=1:3), the HPSEC analysis demonstrated that BSA only can bound to TAT motif of T-GFP (Please see below or new supplementary Fig. 6b and 6c). On the other hand, FACS analysis in the HEK-293T cells showed that the MFI of 5 μ M T-GFP-treated cells in the presence of BSA (50 g/L) decreased to 38.7% of its counterpart in the serum-free DMEM (Please see below

or revised Fig. 2a). Consistent with previous speculations, our data show that the interaction of albumin with TAT contributes to the low serum tolerance of TAT-mediated delivery. As expected, interaction between TL-GFP and BSA was observed (Please see below or new supplementary Fig. 6d) as well, however, MFI of TL-GFP-treated cells in the presence of BSA remained 83.1% of that in the free DMEM (Please see below or revised Fig. 2f), **suggesting that dimerization of TAT could alleviate the binding of BSA-induced inhibitory effect on the endocytosis.** Even though, due to the restricted resolution of SEC analysis in this research, we cannot identify whether both proteins associate only one molecule of albumin. **Therefore, available data are not going to provide a definitive answer to the question if CPP interacts with serum albumin in a monovalent manner, further study needs to be performed to verify this assumption.** Accordingly, we now deleted this speculation as suggested and rephrased this statement (Page 6, Line 142-156; Page 8, Line 172-176).

New supplementary figure. 6 The identification of the binding between TAT motif in the constructs with BSA. a The SDS-PAGE analysis of BSA, and HPSEC analysis of BSA in the different buffer. **b-d** The analysis of interaction between GFP-related proteins and BSA with the molar ratio of 1:3 using HPSEC;

GFP (b), T-GFP (c), TL-GFP (d). e HPSEC retention time of protein molecular weight standards in the buffer of PBS containing 0.4 M NaCl. Uncropped images of PAGE are shown in supplementary Fig.16.

Revised Fig. 2 Dimerization of CPP-fused proteins enhances endocytosis and serum tolerance. a The MFI of T-GFP treated HEK-293T cells (5 μ M) in the free-serum DMEM, DMEM containing BSA (50g/L) or 100% FBS. **f** The MFI of TL-GFP treated HEK-293T cells (5 μ M) in the free-serum DMEM, DMEM containing BSA (50g/L) or 100% FBS. Percentages presented in the columns were obtained by dividing the MFI of treated cells in the given condition by the MFI of treated cells in free-serum DMEM. The data are expressed as means \pm s.e.m: * P<0.05, ** P<0.01, *** P<0.001, **** P < 0.000, and NS, no significant difference.

7. Line 170 ... eTAT showed the highest efficiency for Ppm1b delivery (Fig. 3a and supplementary fig. 9).

Actually neither figure shows delivery efficiency.

Re: Thanks for your valuable suggestion. In the original submission, we speculated the proportion of survival L929 post treatment with the mouse TNF- α and zVAD (TNZ), has positive correlation with the level of Ppm1b in the cytosol(Chen et al. (2015) Nat Cell Biol, 17: 434-444.). Therefore, we only detected PI uptake efficiency of L929 after treatment to evaluate the delivery efficiency. However, as you correctively pointed out, it was not a direct evidence of higher delivery efficiency. To solve this issue, in this revision, we have evaluated the level of Ppm1b in the L929 treated with eTAT-Ppm1b and other constructs directly by western blotting at two key time point by western blotting. As shown in new Fig.3a below, the results of 30 min post treatment could indicate the level of total cellular internalization (cytosolic delivery + endosomal entrapment), while that of 12 h post treatment would indirectly reflect the level of cytosolic delivery (not all the cytosolic delivery), due to the entrapped proteins in the endosomes would be degraded quickly in the lysosome. As shown in revised Fig.3a or see below, at 30 min post treatment, the level of Ppm1b delivered by eTAT in L929 was highest and that of all other TAT-based constructs are about same except Ppm1b alone. When 12 h post incubation onset, the western blotting demonstrated that higher level of Ppm1b (cleaved form) was detected in the cells treated with eTAT-Ppm1b than TINNe-, and no Ppm1b protein could be detected in cells treated with other TAT-based constructs. The efficient cytosolic delivery of eTAT-Ppm1b at 12 h post incubation onset, was attributed to enhanced cellular uptake and more efficient endosomal escape. **In general, these data suggest the eTAT show the highest**

efficiency for Ppm1b delivery. Now, we have integrated above results and corresponding description into the revised MS (Page 10, Line 237-241).

a

Revised Fig. 3 eTAT-Ppm1b suppresses TNF-induced necroptosis *in vitro* and *in vivo*. a Immunoblot analysis of level of Ppm1b in L929 cells treated with Ppm1b-related proteins (1 μ M) at indicated time point.

8. line 210 ... and were intravenously injected with 20 nmol Ppm1b protein twice

Material and Methods says that 5 nmoles (line 451). Which dose was actually used?

Re: We apologize for this error. We confirmed that 5 nmol Ppm1b was used and it has been corrected in the revised MS (Page 13, Line 295).

References

- Aguilera, T. A., E. S. Olson, M. M. Timmers, T. Jiang and R. Y. Tsien (2009). "Systemic *in vivo* distribution of activatable cell penetrating peptides is superior to that of cell penetrating peptides." *Integr Biol (Camb)* **1**(5-6): 371-381.
- Cai, S. R., G. Xu, M. Becker-Hapak, M. Ma, S. F. Dowdy and H. L. McLeod (2006). "The kinetics and tissue distribution of protein transduction in mice." *Eur J Pharm Sci* **27**(4): 311-319.
- Chen, W., J. Wu, L. Li, Z. Zhang, J. Ren, Y. Liang, F. Chen, C. Yang, Z. Zhou, S. S. Su, X. Zheng, Z. Zhang, C. Q. Zhong, H. Wan, M. Xiao, X. Lin, X. H. Feng and J. Han (2015). "Ppm1b negatively regulates necroptosis through dephosphorylating Rip3." *Nat Cell Biol* **17**(4): 434-444.
- Del'Guidice, T., J. P. Lepetit-Stoffaes, L. J. Bordeleau, J. Roberge, V. Theberge, C. Lauvaux, X. Barbeau, J. Trottier, V. Dave, D. C. Roy, B. Gaillet, A. Garnier and D. Guay (2018). "Membrane permeabilizing amphiphilic peptide delivers recombinant transcription factor and CRISPR-Cas9/Cpf1 ribonucleoproteins in hard-to-modify cells." *PLoS One* **13**(4): e0195558.
- Erazo-Oliveras, A., K. Najjar, L. Dayani, T. Y. Wang, G. A. Johnson and J. P. Pellois (2014). "Protein delivery into live cells by incubation with an endosomolytic agent." *Nat Methods* **11**(8): 861-867.
- Evans, B. C., R. B. Fletcher, K. V. Kilchrist, E. A. Dailing, A. J. Mukalel, J. M. Colazo, M. Oliver, J. Cheung-Flynn, C. M. Brophy, J. W. Tierney, J. S. Isenberg, K. D. Hankenson, K. Ghimire, C. Lander, C. A. Gersbach and C. L. Duvall (2019). "An anionic, endosome-escaping polymer to potentiate intracellular delivery of cationic peptides, biomacromolecules, and nanoparticles." *Nat Commun* **10**(1): 5012.

Futaki, S., I. Nakase, T. Suzuki, Z. Youjun and Y. Sugiura (2002). "Translocation of branched-chain arginine peptides through cell membranes: flexibility in the spatial disposition of positive charges in membrane-permeable peptides." Biochemistry **41**(25): 7925-7930.

He, H., L. Sun, J. Ye, E. Liu, S. Chen, Q. Liang, M. C. Shin and V. C. Yang (2016). "Enzyme-triggered, cell penetrating peptide-mediated delivery of anti-tumor agents." J Control Release **240**: 67-76.

Jarver, P., I. Mager and U. Langel (2010). "In vivo biodistribution and efficacy of peptide mediated delivery." Trends Pharmacol Sci **31**(11): 528-535.

Jiang, T., E. S. Olson, Q. T. Nguyen, M. Roy, P. A. Jennings and R. Y. Tsien (2004). "Tumor imaging by means of proteolytic activation of cell-penetrating peptides." Proc Natl Acad Sci U S A **101**(51): 17867-17872.

Kalafatovic, D. and E. Giralt (2017). "Cell-Penetrating Peptides: Design Strategies beyond Primary Structure and Amphipathicity." Molecules **22**(11).

Kauffman, W. B., S. Guha and W. C. Wimley (2018). "Synthetic molecular evolution of hybrid cell penetrating peptides." Nat Commun **9**(1): 2568.

Kawamura, K. S., M. Sung, E. Bolewska-Pedyczak and J. Gariepy (2006). "Probing the impact of valency on the routing of arginine-rich peptides into eukaryotic cells." Biochemistry **45**(4): 1116-1127.

Kosuge, M., T. Takeuchi, I. Nakase, A. T. Jones and S. Futaki (2008). "Cellular internalization and distribution of arginine-rich peptides as a function of extracellular peptide concentration, serum, and plasma membrane associated proteoglycans." Bioconjug Chem **19**(3): 656-664.

Lee, Y. J., G. Johnson, G. C. Peltier and J. P. Pellois (2011). "A HA2-Fusion tag limits the endosomal release of its protein cargo despite causing endosomal lysis." Biochim Biophys Acta **1810**(8): 752-758.

Lo, S. L. and S. Wang (2008). "An endosomolytic Tat peptide produced by incorporation of histidine and cysteine residues as a nonviral vector for DNA transfection." Biomaterials **29**(15): 2408-2414.

Lonn, P. and S. F. Dowdy (2015). "Cationic PTD/ CPP-mediated macromolecular delivery: charging into the cell." Expert Opin Drug Deliv **12**(10): 1627-1636.

Lonn, P., A. D. Kacsinta, X. S. Cui, A. S. Hamil, M. Kaulich, K. Gogoi and S. F. Dowdy (2016). "Enhancing Endosomal Escape for Intracellular Delivery of Macromolecular Biologic Therapeutics." Sci Rep **6**: 32301.

Milech, N., B. A. Longville, P. T. Cunningham, M. N. Scobie, H. M. Bogdawa, S. Winslow, M. Anastasas, T. Connor, F. Ong, S. R. Stone, M. Kerfoot, T. Heinrich, K. M. Kroeger, Y. F. Tan, K. Hoffmann, W. R. Thomas, P. M. Watt and R. M. Hopkins (2015). "GFP-complementation assay to detect functional CPP and protein delivery into living cells." Sci Rep **5**: 18329.

Mitchell, D. J., D. T. Kim, L. Steinman, C. G. Fathman and J. B. Rothbard (2000). "Polyarginine enters cells more efficiently than other polycationic homopolymers." J Pept Res **56**(5): 318-325.

Mohammed, A. F., A. Abdul-Wahid, E. H. Huang, E. Bolewska-Pedyczak, M. Cydzik, A. E. Broad and J. Gariepy (2012). "The *Pseudomonas aeruginosa* exotoxin A

translocation domain facilitates the routing of CPP-protein cargos to the cytosol of eukaryotic cells." J Control Release **164**(1): 58-64.

Mueller, J., I. Kretzschmar, R. Volkmer and P. Boisguerin (2008). "Comparison of cellular uptake using 22 CPPs in 4 different cell lines." Bioconjug Chem **19**(12): 2363-2374.

Neundorff, I., R. Rennert, J. Hoyer, F. Schramm, K. Lobner, I. Kitanovic and S. Wolfli (2009). "Fusion of a Short HA2-Derived Peptide Sequence to Cell-Penetrating Peptides Improves Cytosolic Uptake, but Enhances Cytotoxic Activity." Pharmaceuticals (Basel) **2**(2): 49-65.

Patel, S. G., E. J. Sayers, L. He, R. Narayan, T. L. Williams, E. M. Mills, R. K. Allemann, L. Y. P. Luk, A. T. Jones and Y. H. Tsai (2019). "Cell-penetrating peptide sequence and modification dependent uptake and subcellular distribution of green fluorescent protein in different cell lines." Sci Rep **9**(1): 6298.

Sudo, K., K. Niikura, K. Iwaki, S. Kohyama, K. Fujiwara and N. Doi (2017). "Human-derived fusogenic peptides for the intracellular delivery of proteins." J Control Release **255**: 1-11.

Sung, M., G. M. Poon and J. Garipey (2006). "The importance of valency in enhancing the import and cell routing potential of protein transduction domain-containing molecules." Biochim Biophys Acta **1758**(3): 355-363.

Reviewers' Comments:

Reviewer #2:

Remarks to the Author:

The authors have addressed the referee's critics properly. The manuscript is ready for publication.

Reviewer #4:

None

Authors' responses to reviewers' comments for the manuscript (NCOMMS-20-45381A):

Efficient intracellular delivery of proteins by a multifunctional chimaeric peptide both *in vitro* and *in vivo*

Siyuan Yu¹, Han Yang¹, Tingdong Li¹, Haifeng Pan¹, Shuling Ren, Guoxing Luo, Jinlu Jiang, Linqi Yu, Binbing Chen, Yali Zhang, Shaojuan Wang, Rui Tian, Tianying Zhang, Shiyin Zhang, Yixin Chen, Quan Yuan*, Shengxiang Ge*, Jun Zhang, and Ningshao Xia*

Our revisions are described below in a point-by-point manner. For easier reading, the reviewers' comments are marked **in black**, responses are marked **in blue**, added sentences in the main text are marked **in red**. Questions are copy-pasted from the *Nature Communications* Editor's decision correspondence. Changes in the main text of the revised manuscript are highlighted **in yellow**.

Reviewer #2 (Remarks to the Author):

The authors have addressed the referee's critics properly. The manuscript is ready for publication.

Re: Again, thanks for your thoughtful comments and ideas, which helped us to significantly increase the impact of our work.

Reviewer #4 (Remarks to the Author):

We would like to thank the Reviewer #4 for your positive assessment of our revision. Following your advice, we have made a few changes to our manuscript, as highlighted in the new version of the text; a point-by-point response to your comments is provided below:

Question 1 of reviewer #3:

The authors have included the recommended references of Futaki's (15) or Tsien's work (45-46). Furthermore, they have included in the revised manuscript a clear explanation of the design of their CPP-based intracellular protein delivery system (eTAT, enhanced TAT).

Re: Thank you for these comments.

Question 2 of reviewer #3:

Questions around the His-tag used for protein purification which could have an effect as proton sponge to facilitate endosomal escape were answered with the help of the presented Figure R2. The results are conclusive but not mentioned in the revised manuscript. However, as highlighted by reviewer #3 a lot of work was performed on poly-His grafted CPP sequences in order to facilitate endosomal escape. In the case of the 6X-His protein-tag, the proton sponge effect is not visible showing no increase in relative MFI. **I believe that this information should be included in the manuscript.**

Re: Thank you for these constructive comments. As you suggested, a new supplementary figure.6 (original Fig.R2 in rebuttal letter) and corresponding description have been included in the revised manuscript. **The following sentences were included in revised manuscript (Line 126-135):**

"Notably, poly-histidine tag (6×His-tag) was presented in all of GFP₁₋₁₀-NLS-related recombinant proteins to facilitate the purification of these proteins. Histidine residues are known to serve as a proton sponge and facilitate escape from endosomes, thus poly-histidine sequences have been used as motifs to improve endosomal escape in trans-delivery (co-incubation) of gene (Lo and Wang (2008) Biomaterials, 29: 2408-2414.) or ribonucleoproteins (Del'Guidice et al. (2018) PLoS One, 13: e0195558.) previously. To evaluate the impact of 6×His-tag on endosomal escape when fused to CPP-cargo proteins, the HEK-293T-GFP₁₁ cells were treated by 5 μM TINNe-GFP₁₋₁₀-NLS with 6×His-tag (6H+) or not (6H-) separately. Our data show that there is no significant difference between the MFI of HEK-293T-GFP₁₁ treated by the two proteins, suggesting that proton sponge effect of His-tag is not obvious when fused to CPP-cargo proteins (Supplementary Fig. 6)."

Question 3 of reviewer #3:

Addition of INF7 to Tat reduced the amount of positive charges but increase internalization (Fig 1g and h). The authors argued that this result could be explained by those obtained

by Neundorf, Rennert et al 2009 showing conformational changes of the conjugated sepeptide which might promote cell membrane interaction.

I have a doubt on this explanation. Tat is able to internalize big proteins as well as small peptides. **I would rather believe that the INF7 peptide acts as spacer or linker making Tat more accessible for the interaction with the cell surface GAGs.** If the authors believe on the conformational change of Tat versus Tat-INF7 – both peptides could be analyzed by circular dichroism in order to see changes and to confirm their hypothesis.

Re: Thank you for these helpful comments. Firstly, we completely agreed with you that our observation could be attributed to that INF7 peptide acts as spacer or linker making TAT more accessible for the interaction with the cell surface GAGs, which would be one of the possible mechanisms behind this observation. Secondly, with regard to our explanation for the conformational change of TAT in previous revision, due to their data (Neundorf et al. (2009) *Pharmaceuticals (Basel)*, 2: 49-65.) was obtained by investigating the difference of cellular uptake of carboxyfluorescein (CF)-labelled TAT or TAT-PMAP, accordingly, we indicated that it was a one of probable mechanism behind this observation although we did not know whether this conformational change occurs when fused to cargo proteins. With regard to your suggested experiments that both peptides (TAT, TAT-INF7) could be analyzed by circular dichroism to see changes and to confirm this hypothesis. We have discussed with the experts doing active research in protein structure, and they suggested that the molecular size of TAT (13 aa) was too small in comparison with its fused-cargo proteins (ie,GFP, 238 aa), and it was difficult to accurately identify whether the conformational change of TAT appear in the TAT-INF7-cargo proteins by using this approach. **Even though, based on these comments, in this revision, your suggestion, and our original explanation, were clearly described as two potential mechanisms behind this observation, and we believed this could be addressed in the future studies. The following sentences were included in revised manuscript (Line 354-359):**

*"The increases in cytosolic uptake by fusion of PMAP (HA2) to CPP has been also observed by Wöfl group and they proposed that the enhancement was due to conformational changes(Neundorf, Rennert et al. (2009) *Pharmaceuticals (Basel)*, 2: 49-65.), which may enhance the membrane interaction or endosome disruption. Besides, the INF7 peptide could also act as a spacer or linker, which may make TAT more accessible for the interaction with the cell surface components (ie, GAGs), and thus increase the intracellular uptake."*

Question 4 of reviewer #3:

I agree completely with the response of the authors. Indeed, in particular cases protein dimerization could be observed in SDS-PAGE even under denaturing conditions.

Re: We appreciated you for these positive comments.

Question 5 of reviewer #3:

eTAT-Ppm1b distribution in the caecum – as required by reviewer #3, the authors have commented this fact in the result and discussion sections.

Re: Thank you for these comments.

Question 6 of reviewer #3:

Authors have corrected that Tat interacts with GAGs, protein receptors or phospholipids but not directly with the membrane.

Re: Thank you for these comments.

Questions 7-9 of reviewer #3:

The authors have focused on their results in the discussion, have added references and have reviewed the manuscript.

Re: We appreciated you for these comments.

Specific comments 1, 3, 4, 5, 7, 8 of reviewer #3 were answered by the authors.

Re: Thank you for these comments.

Specific comment 2 of reviewer #3 concerning the used doses was answered by the authors by introducing a new supplementary figure 16.

Re: Thank you for these comments.

Specific comment 6 of reviewer #3 concerning the BSA “resistance” of the dimeric protein construct was answered and discussed with a new supplementary figure 6 showing that Tat dimers could alleviate its endocytosis-dependent internalization inhibited by BSA. Please check distances between title and graphic of revised Fig. 2a and 2f.

Re: Thank you for helpful comments. We have revised the position of title of Fig.2a and 2f as you suggested.

After these minor modifications, the manuscript could be published in Nature Communications.

Re: Again, many thanks for your encouraging comments.

References

Del'Guidice, T., J. P. Lepetit-Stoffaes, L. J. Bordeleau, J. Roberge, V. Theberge, C. Lauvaux, X. Barbeau, J. Trottier, V. Dave, D. C. Roy, B. Gaillet, A. Garnier and D. Guay (2018). "Membrane permeabilizing amphiphilic peptide delivers recombinant transcription factor and CRISPR-Cas9/Cpf1 ribonucleoproteins in hard-to-modify cells." *PLoS One* **13**(4): e0195558.

Lo, S. L. and S. Wang (2008). "An endosomolytic Tat peptide produced by incorporation of histidine and cysteine residues as a nonviral vector for DNA transfection." *Biomaterials* **29**(15): 2408-2414.

Neundorf, I., R. Rennert, J. Hoyer, F. Schramm, K. Lobner, I. Kitanovic and S. Wolfli (2009). "Fusion of a Short HA2-Derived Peptide Sequence to Cell-Penetrating Peptides Improves Cytosolic Uptake, but Enhances Cytotoxic Activity." *Pharmaceuticals (Basel)* **2**(2): 49-65.

Reviewers' Comments:

Reviewer #4:

Remarks to the Author:

NCOMMS-20-45381A

Efficient intracellular delivery of proteins by a multifunctional chimaeric peptide both in vitro and in vivo

Siyuan Yu, Han Yang, Tingdong Li, Haifeng Pan, Shuling Ren, Guoxing Luo, Jinlu Jiang, Linqi Yu, Binbing Chen, Yali Zhang, Shaojuan Wang, Rui Tian, Tianying Zhang, Shiyin Zhang, Yixin Chen, Quan Yuan*, Shengxiang Ge*, Jun Zhang, and Ningshao Xia*

After having carefully read the second round of correction of the authors, I recommande this manuscript for publication at Nature Communication.

Prisca Boisguerin

Authors' responses to **reviewers' comments** for the manuscript (NCOMMS-20-45381B):

Efficient intracellular delivery of proteins by a multifunctional chimaeric peptide *in vitro* and *in vivo*

Siyuan Yu, Han Yang, Tingdong Li, Haifeng Pan, Shuling Ren, Guoxing Luo, Jinlu Jiang, Linqi Yu, Binbing Chen, Yali Zhang, Shaojuan Wang, Rui Tian, Tianying Zhang, Shiyin Zhang, Yixin Chen, Quan Yuan*, Shengxiang Ge*, Jun Zhang, and Ningshao Xia*

Our revisions are described below in a point-by-point manner. For easier reading, the reviewers' comments are marked **in black**, responses are marked **in blue**. Questions are copy-pasted from the *Nature Communications* Editor's decision correspondence.

Reviewer #4 (Remarks to the Author):

After having carefully read the second round of correction of the authors, I recommande this manuscript for publication at Nature Communication.

Prisca Boisguerin

Re: Thank you very much for assessing our revision, and for the helpful suggestions during review.